# A LONG RANGE FOUNDATION MODEL FOR ZERO-SHOT PREDICTIONS IN SINGLE-CELL AND SPATIAL TRANSCRIPTOMICS DATA

## ABSTRACT

Large transformers pretrained with language model objectives have demonstrated success in multiple fields, and have tremendous potential for modeling single-cell RNA-seq and spatial transcriptomics data. However, these approaches are yet to overcome various challenges, including inductive biases that hinder generalization, artifacts and quality of the underlying data, as well as downstream evaluation pipelines that do not reflect the biological challenges in the field. In this work, we propose a new framework, sCellTransformer (sCT), that relies on a first principles formulation of the problem as well as a validation pipeline designed to evaluate models generalization through zero-shot predictions. sCT leverages a long-range convolutional-transformer architecture that is trained from unprocessed single-cell and spatial transcriptomics data. In contrast to previous works, sCT represents cells with up to 20,000 protein-coding genes, processes sets of multiple cells, and predicts about a million discretized gene expression tokens. We show that representing gene expression as discrete levels allows us to mitigate the high sparsity present in single-cell data both during training and evaluation. We present state-of-the-art empirical results on several zero-shot gene expression imputation, cell-typing, and clustering tasks in both single-cell as well as spatial domains, outperforming current foundation models.

## 1 INTRODUCTION

Assays measuring gene transcription, such as single-cell RNA-sequencing (scRNA-seq) and spatial transcriptomics (ST), have become indispensable tools in biology. These next-generation sequencing (NGS) technologies provide high-resolution insights into cellular mechanisms by analyzing RNA expression within individual cells or localized cell populations in tissues. ScRNA-seq has proven invaluable for assessing tumors at the genetic level (De Falco et al., 2023; Dohmen et al., 2022), identifying rare cell types (Jindal et al., 2018), and characterizing gene regulation across tissues (Kartha et al., 2022). ST extends scRNA-seq by incorporating positional information alongside gene expression, facilitating more precise modeling of cellular interactions. The widespread adoption of these assays has led to a surge in publicly available, high-quality sequencing data, creating a demand for approaches that can effectively analyze and interpret this data.

This need has fueled the development of deep learning-based methods (Cui et al., 2024; Wen et al., 2023; Schaar et al., 2024; Yang et al., 2022; Rosen et al., 2023; Lopez et al., 2018; 2019; Lotfollahi et al., 2019) aimed at learning transferable, contextual representations of single-cell data. These methods tackle complex downstream tasks like cell-type annotation, data integration, and gene expression imputation (Lähnemann et al., 2020). More recently, self-supervised models trained with language modeling objectives, such as BERT (Devlin, 2018) and GPT (Brown, 2020), have been adapted to this domain. These approaches generally treat gene sequences within individual cells as input sequences, incorporating gene identifiers and their corresponding expression levels. Masked language modeling or autoregressive modeling is then employed to predict gene expressions or gene IDs. Models like scGPT (Cui et al., 2024), scBERT (Yang et al., 2022), and CellPLM (Wen et al., 2023) exemplify this framework for learning cell-level embeddings.

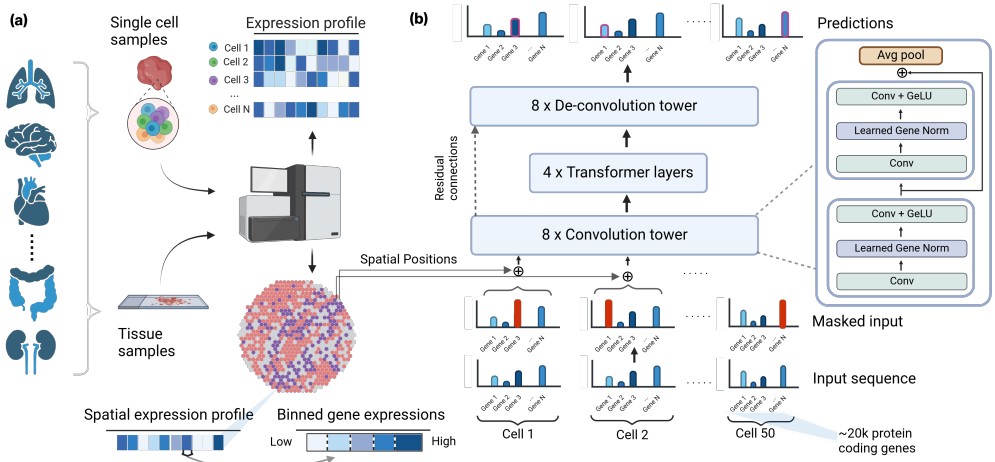

Figure 1: **sCellTransformer : Framework**. sCT leverages scRNA-seq data to construct sequences of gene expression values coming from multiple cells within the same sample. These sequences contain approximately one million tokens (batches of 50 cells with ∼20,000 gene expression values each) and are fed into a convolution tower that compresses them, followed by a multi-head attention block and a deconvolution tower that recovers the original sequence through gene expression level predictions. It also incorporates spatial coordinates for each cell as positional embeddings. sCT allows for imputing gene expression levels both for specific genes, and whole cells.

However, the Transformer architecture, while capable of capturing non-local patterns, is constrained by its quadratic memory scaling with respect to input length. The vast number of genes in a cell (around 20,000 coding genes) limits models like scBERT and scGPT to processing a single cell at a time, often requiring gene subset selection. CellPLM introduced the concept of "bags-of-cells," demonstrating the benefits of encoding multiple cells simultaneously. However, their gene expression embedder, which sums gene embeddings across cells, can lead to information loss. Moreover, these models typically predict continuous gene expression values, neglecting the inherent noise and sparsity of scRNA-seq data.

Another challenge is gene-level dropout, a consequence of capturing only a fraction of expressed mRNA (Haque et al., 2017), and is a prevalent noise source in scRNA-seq data, with expression estimates that vary significantly depending on experimental conditions. Single-cell data is also inherently sparse (Jiang et al., 2022). For example, in the CxG dataset (CZI Single-Cell Biology et al., 2023), on average, less than 6% of protein-coding genes are expressed in a cell. Standard regression losses like mean squared error (MSE) do not effectively account for this highly skewed data. Additionally, ST technology, being array-based, samples only a limited subset of cells within a tissue, resulting in missing gene expression data for cells outside the array spots. We believe future models should be evaluated on their ability to predict these missing values.

Furthermore, current benchmarks for evaluating single-cell models often lack a focus on generalization. ScRNA-seq analysis is largely exploratory, often involving clustering latent cell vectors for cell typing without predefined labels (Hie et al., 2020; Argelaguet et al., 2021; Heumos et al., 2023). However, models like scGPT, geneformer, scBERT, and CellPLM require fine-tuning on new datasets for optimal performance (Kedzierska et al., 2023). This highlights a discrepancy between the field's need for exploratory analysis and the current limitations of single-cell foundation models. Therefore, we advocate for evaluating these models on their zero-shot performance, which we believe is crucial for bridging this gap and advancing the field. This aligns with similar progress towards zero-shot foundation models in protein (Truong Jr & Bepler, 2023) and genomics research (Nguyen et al., 2024).

To address these limitations, we introduce sCT , a novel architecture based on a convolutional-Transformer design that can process up to 1 million gene expression values per input. This enables sCT to process up to 50 cells simultaneously, encompassing all 20,000 protein-coding genes. Instead of continuous predictions, sCT predicts discrete gene expression values over a fixed number of

levels. This approach mitigates measurement noise and accounts for data sparsity. Notably, we emphasize the use of the Matthews correlation coefficient (MCC) (Matthews, 1975), which is robust to label imbalance (Chicco & Jurman, 2020). We also introduce a new benchmark focused on evaluating zero-shot capabilities in new data domains, revealing the limitations of existing methods like scGPT in this setting. Our sCT architecture effectively addresses this gap and demonstrates significantly improved zero-shot performance.

Formally, our contributions are as follows:

1. We present the first single-cell model architecture in transcriptomics capable of processing up to 1 million gene expression values (tokens) as input, far exceeding the capacity of existing Transformer-based models. sCT can be trained on both single-cell and spatial transcriptomics data simultaneously.
2. We propose a shift in how single-cell models are evaluated in transcriptomics by introducing a new benchmark focused on zero-shot performance in new domains. We also advocate for using metrics robust to highly imbalanced datasets, demonstrating the shortcomings of current state-of-the-art models in such settings.
3. We show that our architecture overcomes some limitations of current single-cell models for transcriptomics, achieving substantially improved zero-shot performance in new domains.
4. We provide a comprehensive ablation study to support our design choices and validate the model. We believe the architectural choices presented establish best practices for future research in this field.

## 2   RELATED WORK

Prior to the surge in recent developments for pretrained foundation models, deep learning approaches such as variational auto-encoders (Bereket & Karaletsos, 2023; Roohani et al., 2024), supervised or training (Lotfollahi et al., 2019; Lopez et al., 2018), and semi-supervised training (Xu et al., 2021; 2024) were successful at solving specific tasks in transcriptomics. More recently, advances in large Transformer-based models (Vaswani et al., 2017), enabled the training of foundation models where a single model can solve a wide range of tasks in computational biology, specifically transcriptomics.

**Self-supervised models for scRNA-seq**   ScBERT (Yang et al., 2022) and scGPT (Cui et al., 2024) leverage BERT-style (Devlin, 2018) masked language modeling and next-token prediction (Brown, 2020) respectively on gene expression sequences for representation learning. XtrimoGene (Gong et al., 2023) and scFoundation (Hao et al., 2024a) build upon these by adding a learnable discretization layer. Nicheformer (Schaar et al., 2024) also leverages BERT-style training by carefully curating scRNA-seq data and extending sequence tokenization to include metadata for species, tissue, and sequencing assay, as well as leveraging spatial transcriptomics datasets and assay-specific gene expression bias terms. Models like Levine et al. (2024) and Theodoris et al. (2023) use objectives like next-token prediction or masked language modeling respectively on rank-ordered gene identifiers to learn useful representations. We point the readers to Heydari & Sindi (2023) for a thorough survey of recent methods in the field.

**Spatial Transcriptomics (ST)**   Spatial transcriptomics data is often integrated alongside scRNA-seq (Lopez et al., 2019), or histopathological images (Jaume et al., 2024) to model cellular interactions in tissues. Typically, this is done through supervised training (Biancalani et al., 2021) or contrastive learning (Li et al., 2023). SpaGE (Abdelaal et al., 2020) focused on deconvolving such sequences into identifiable mixtures of celltypes for interpretation. Other approaches focus more on representation learning through graph neural networks (Ma et al., 2024), self-supervised learning (Xu et al., 2024), or dictionary learning (Hao et al., 2024b). However, these approaches often require matched multimodal data, or do not generalize without being finetuned. CellPLM (Wen et al., 2023) improves upon these approaches by training a transformer-based model on collections of cell, allowing their model to leverage the mutual information between proximal cells in situ.

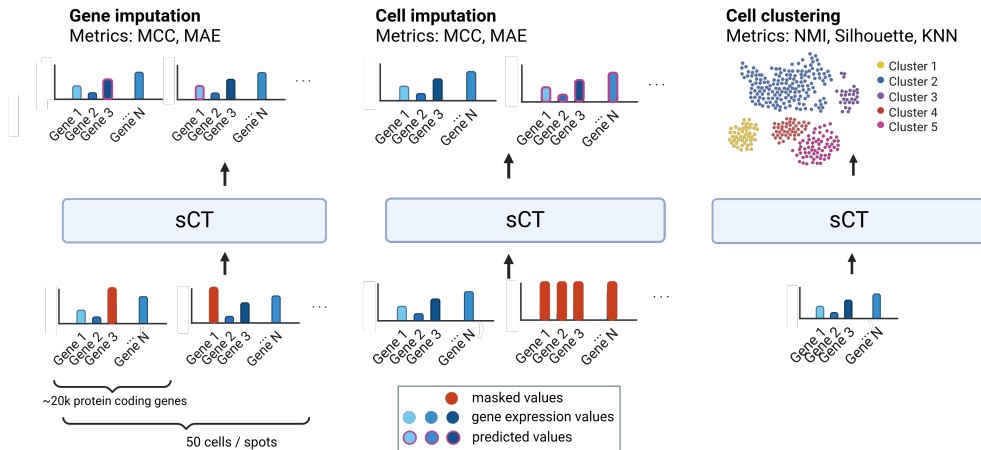

Figure 2: **Our benchmark to evaluate generalization capabilities.** We consider three zero-shot tasks, all performed on data domains that were not present during training. (1) Gene imputation reconstruction, where we randomly mask a fraction of the tokens of each cell and the goal is to reconstruct them; (2) cell imputation reconstruction, where all tokens from the same cell are masked and the goal is to reconstruct them based on neighboring cells. Note that, for both tasks (1) and (2) the model takes 50 cells as input. Finally, (3) cell clustering, where the embeddings obtained by sCT are used to do zero-shot cell type annotation and clustering.

## 3   SCELLTRANSFORMER (SCT)

In this section, we present sCT , a model for single-cell and spatial transcriptomics data, trained through masked language modeling. We present a series of important choices in the architecture that led to significant performance improvements over competitor models.

**Data processing.**   Similarly to other transcriptomics pipelines (Wen et al., 2023; Yang et al., 2022), we select and keep $m$ known protein-coding gene identifiers from Ensembl (Martin et al., 2022), with $m \approx 20,000$. The positions of these genes across all the different cells are fixed for constructing the input sequence. For every study, we also zero out gene expression values for genes where the total count is less than $0.003$ times the number of total cells, as these are considered to be noisy measurements (Cui et al., 2024). We apply log transformation on the raw gene expressions and then bin the non-zero gene expressions for every cell using a uniform binning on the range of non-zero gene expression values. Gene expression values are therefore encoded into discrete levels. Note that all non-expressed or unmeasured genes are encoded as the zero bin, and included in the representation. We do not use any other preprocessing operations, in an effort to reduce biases.

**Input representation.**   As gene expression values are binned, we represent them as tokens with a vocabulary size equaling the number of bins. Embeddings are learned to represent each token. To allow the model to exploit intercellular redundancies and learn co-expressions, we stack $k$ processed cells into a single input sequence. The final input sequence can then be represented as a sequence of $k \times m$ gene expression levels (see Fig. 1 for reference). Note that, for scRNA-seq, the sequence is constructed by sampling cells from the same study, while for spatial transcriptomics, by sampling within the neighborhood of a cell. Motivated by recent works Wen et al. (2023), we use $k = 50$ cells at training time. Each input sequence thus consists of approximately $10^6$ tokens of binned gene expression values.

**Convolutional-transformer architecture.**   To address the quadratic scaling of self-attention with input length, which prohibits processing $10^6$ tokens with standard transformers, we propose a convolutional-transformer architecture. Inspired by the UNet architecture (Ronneberger et al., 2015) and its adaptation for biological sequences (Linder et al., 2023), our model employs a convolutional tower to compress the input via consecutive 1D convolution and max pooling layers. This significantly reduces computational cost while preserving local gene expression patterns. The resulting

compressed embeddings are then processed by a series of standard Transformer blocks (Vaswani et al., 2017) to capture global interactions. This bottleneck layer maintains cellular context, crucial for accurately modeling scRNA-seq data. Finally, a deconvolutional block, utilizing residual connections, upsamples the embeddings to the original input length. The imputation head then predicts gene expression from these learned embeddings. While using convolutions for unordered data like scRNA-seq might seem counterintuitive, our primary motivation was dimensionality reduction to allow for input of multiple cells. The convolutional architecture presents an efficient mechanism to achieve this, while also outperforming baselines as seen in our results below.

**Shared Layer Norm.** We apply layer norm (Lei Ba et al., 2016) with a learned bias and scale shared across gene positions for every cell. Contrary to the standard layer norm, which learns a common normalization bias and scales across all tokens, here we assign learnable biases and scales to every single gene position in a cell. These parameters are therefore repeated $k$ times where $k$ is the number of cells per sequence. Intuitively, these layers learn gene level statistics across the cells, and over the training dataset. We show empirically that these normalization layers play a significant role in improving gene expression predictions in our ablation studies.

**Positional Embeddings.** Standard Transformer models add positional encoding to their input data. Here, as there is no natural order neither in the cells nor in the genes, we removed this encoding. Instead, for single-cell data we replace it by a simple constant encoding per cell that is broadcast to all the genes within the cell. In the case of spatial transcriptomics data, this constant encoding is replaced by a 2D aware sinusoidal position encoding (SPE) that represents the relative positions of cells within the FOV. Similar to Klemmer et al. (2023), we construct them to ensure scale invariance across fields of view and experiments, see Eq. 1.

$$ST(\boldsymbol{C}, \sigma_{min}, \sigma_{max}) = \left[\cos\left(\frac{\boldsymbol{C}^{[v]}}{\sigma_{min}g^{s/(S-1)}}\right); \sin\left(\frac{\boldsymbol{C}^{[v]}}{\sigma_{min}g^{s/(S-1)}}\right)\right], \quad s \in \{0, ..., S-1\}, v \in \{1, 2\}$$
(1)

Where $\boldsymbol{C}$ are the spatial coordinates, $S$ is the total number of scales used, $\sigma_{min}$ and $\sigma_{max}$ are respectively the minimum and maximum scales, and $g = \sigma_{max}/\sigma_{min}$. Note that for single cell inputs that have no relative position per cell, the position embeddings are hard-coded to zero.

**Stratified masking procedure.** To address data imbalance, sCT employs a stratified masking approach during masked language modeling (MLM) training (Devlin, 2018). Instead of uniform masking, non-expressed genes are masked with a 1% probability, while expressed genes are masked at 15%. These ratios, determined from training data statistics, ensure a balanced number of masked tokens across expression levels. This strategy effectively counters the bias introduced by imbalanced expression levels, as demonstrated by improved performance in ablations in Sec. 4.

**Loss function.** In opposition to competitor models (Wen et al., 2023; Cui et al., 2024; Yang et al., 2022), sCT predicts probabilities over a vocabulary of tokens instead of continuous values. We thus replace the mean squared error loss by a cross-entropy loss.

**2-stage training.** sCT is trained in two phases using a masked language modeling objective. First, the model is trained on single-cell RNA-sequencing (scRNA-seq) data. We utilize the Cell x Gene (CxG) API to collect all human studies containing both normal and diseased cells, splitting them into training and testing sets at the study level. This yields a training set of 42 million cells. Second, training continues with spatial transcriptomics data from HEST-1K (Jaume et al., 2024), which encompasses data across diverse disease types, tissues, and acquisition methods. We focus solely on human transcriptomics studies within HEST-1K and harmonize the gene identifiers with those used in the scRNA-seq data, resulting in 0.69 million cells for training. This scale is comparable to that used in CellPLM (Wen et al., 2023). Our models is trained on a single NVIDIA A100 GPU for five days. Evaluations are conducted on the same hardware and take approximately 20 minutes. The model[1] has approximately 80 million parameters, a size comparable to CellPLM and scGPT.

---

[1]Model checkpoints and code will be released post-review.

Table 1: **Gene imputation for different masking ratios using scRNA-seq and spatial data.** We compare sCT with literature baselines when masking a fixed fraction of genes for all cells in the input sequence during inference. Note that we only use a fixed stratified masking strategy during training. MCC = Matthews Correlation Coefficient. MAE = Mean Absolute Error. Bold font indicates that one or several algorithms are statistically better than the rest, over 5 evaluations runs. Note that no ST here refers to the base sCT pretrained on scRNA-seq data without Spatial reTraining.

| | Masking ratios | | | | | |
| | 15% | | 30% | | 80% | |
| Model | MCC (↑) | MAE (↓) | MCC (↑) | MAE (↓) | MCC (↑) | MAE (↓) |
| **scRNA-seq** | | | | | | |
| sCT (zero-shot) | **0.49** ±0.01 | **2.00** ±0.04 | **0.47** ±0.02 | **2.00** ±0.05 | **0.37** ±0.01 | **2.31** ±0.06 |
| CellPLM (zero-shot) | **0.49** ±0.02 | 2.24 ±0.05 | 0.45 ±0.02 | 2.38 ±0.05 | 0.15 ±0.02 | 3.30 ±0.08 |
| scGPT (zero-shot) | 0.00 ±0.001 | 260.33 ±70.05 | 0.00 ±0.001 | 266.95 ±81.53 | 0.00 ±0.002 | 268.46 ±92.31 |
| scBERT (zero-shot) | 0.04 ±0.01 | 76.59 ±14.32 | 0.04 ±0.002 | 76.64 ±12.86 | 0.02 ±0.01 | 76.98 ±11.29 |
| MAGIC (fitted) | 0.42 ±0.02 | 2.43 ±0.37 | 0.39 ±0.03 ± | 2.67 ±0.39 | 0.20 ±0.02 | 3.60 ±0.47 |
| **Spatial Transcriptomics (ST)** | | | | | | |
| sCT (sc only) (zero-shot) | 0.05 ±0.01 | 1.40 ±0.05 | 0.05 ±0.01 | 1.41 ±0.05 | 0.03 ±0.01 | 1.51 ±0.05 |
| sCT (sc + ST) (zero-shot) | **0.35** ±0.03 | **1.31** ±0.05 | **0.34** ±0.02 | **1.32** ±0.05 | **0.28** ±0.02 | **1.45** ±0.06 |
| CellPLM (zero-shot) | 0.23±0.02 | 1.48±0.05 | 0.20 ±0.02 | 1.52±0.05 | 0.03 ±0.01 | 2.02 ±0.07 |

## 4 EXPERIMENTS AND RESULTS

As motivated above, we introduce a benchmark focused on evaluating generalization capabilities of transcriptomics self-supervised models to data domains that were not present during training. As such, we study three zero-shot tasks (1) gene imputation, (2) whole cell imputation, and (3) cell embedding clustering and compare sCT to scRNA-seq model competitors and standard bioinformatics baselines. See Fig. 2 for more details about the tasks. Finally, we provide an ablation study to validate our different architectural choices.

**Evaluation Data.** To evaluate model performance, we curated six datasets each for the single-cell and spatial transcriptomics domains. All tasks are performed on each dataset, except for whole-cell masking, which is applicable only to the spatial transcriptomics domain. For single-cell RNA sequencing (scRNA-seq) evaluation, we construct a test set comprising held-out studies from Cell x Gene (CxG) spanning six different tissues. Each study includes a diverse set of cell types, encompassing both normal and diseased cells. We apply the same preprocessing steps used for training. To ensure fair comparison, we maximize the overlap of gene identifiers between the token vocabularies of all models and the test sets. For spatial transcriptomics evaluation, we utilize six held-out fields of view (FOVs) from the HEST dataset, each corresponding to a distinct tissue type. Again, we follow the same preprocessing steps as described earlier. The datasets cover the following tissues: lymph node, colon, lung, kidney, brain, and rectum. Three studies consist of normal cells, two consist of cancerous cells, and one consists of treated cells.

**Baselines.** We compare our models with CellPLM (Wen et al., 2023), scGPT (Cui et al., 2024), and scBERT (Yang et al., 2022). To the best of our knowledge, CellPLM is currently the only other model that can naturally handle both scRNA-seq and spatial data and that also uses a similar approach of ingesting multiple cells at the same time. In the case of the cell embedding clustering task (third task in Fig. 2), we also add Geneformer (Theodoris et al., 2023) to the list of baselines. We have also added several strong bioinformatic baselines for gene imputation and cell clustering; scVI (Gayoso et al., 2022), and scanpy (Wolf et al., 2018) with logistic regression, and $k$-nearest neighbors for cell-typing, and MAGIC (van Dijk et al., 2018) for gene imputation.

**Metrics.** The Matthews Correlation Coefficient (MCC) and the Mean Absolute Error (MAE) are used as evaluation metrics for the imputation tasks. They are known for their robustness to extreme data imbalance and sparsity. Note that we also evaluate all models only on the expressed genes to ensure fairness, as some of the other approaches do not consider non-expressed genes for the

Table 2: **Cell imputation for different numbers of masked cells using both scRNA-seq and spatial transcriptomics data.** We compare sCT with CellPLM when masking all gene expression values (100% masking) for a given number of cells during inference. Predictions are based only on scRNA-seq data. MCC = Matthews Correlation Coefficient (higher is better). MAE = Mean Absolute Error (lower is better).

| | | | Number of masked cells | | | |
| | 1 | | 10 | | 40 | |
| Model | MCC ($\uparrow$) | MAE ($\downarrow$) | MCC ($\uparrow$) | MAE ($\downarrow$) | MCC ($\uparrow$) | MAE ($\downarrow$) |
| **scRNA-seq** | | | | | | |
| sCT (zero-shot) | **0.70** $\pm$0.04 | **1.64** $\pm$0.06 | **0.71** $\pm$0.04 | **1.85** $\pm$0.06 | **0.43** $\pm$0.02 | **2.53** $\pm$0.08 |
| CellPLM (zero-shot) | 0.00 $\pm$0.01 | 3.78 $\pm$0.08 | 0.00 $\pm$0.01 | 3.78 $\pm$0.08 | 0.00 $\pm$0.01 | 3.78 $\pm$0.08 |
| CellPLM$^+$ (zero-shot) | 0.09 $\pm$0.02 | 3.54 $\pm$0.07 | 0.06 $\pm$0.02 | 3.64 $\pm$0.07 | 0.02 $\pm$0.01 | 3.55 $\pm$0.07 |
| $k$-NN smoothing (fitted) | 0.11 $\pm$0.02 | 2.80 $\pm$0.06 | 0.05 $\pm$0.01 | 3.03 $\pm$0.06 | 0.06 $\pm$0.01 | 3.51 $\pm$0.09 |
| **Spatial Transcriptomics (ST)** | | | | | | |
| sCT (zero-shot) | **0.57** $\pm$0.03 | **1.25** $\pm$0.05 | **0.54** $\pm$0.03 | **1.26** $\pm$0.05 | **0.32** $\pm$0.02 | **1.36** $\pm$0.05 |
| CellPLM (zero-shot) | 0.00 $\pm$0.00 | 2.13 $\pm$0.06 | 0.00 $\pm$0.00 | 2.15 $\pm$0.06 | 0.00 $\pm$0.00 | 2.16 $\pm$0.06 |
| CellPLM$^+$ (zero-shot) | 0.13 $\pm$0.02 | 2.12 $\pm$0.06 | 0.13 $\pm$0.02 | 2.10 $\pm$0.06 | 0.12 $\pm$0.02 | 1.84 $\pm$0.05 |
| $k$-NN smoothing (fitted) | 0.00 $\pm$0.01 | 1.87 $\pm$0.05 | 0.01 $\pm$0.01 | 1.92 $\pm$0.05 | 0.03 $\pm$0.01 | 2.18 $\pm$0.06 |

imputation tasks. We also report the MAE metric on raw counts by reversing the log-transform operation wherever appropriate to compare all models on the same range of gene expression values.

For the clustering task, we use zero-shot cell type classification accuracy (KNN), normalized mutual information (NMI), and the Adjusted Rand Index (ARI). For metrics that require raw gene expression counts (such as MAE for example), we transform sCT 's predicted distribution over discrete tokens to continuous values by taking the weighted linear combination of bin medians with the respective probabilities.

**Zero-shot Gene Imputation Results.** Gene expression imputation, illustrated in Fig. 2, is a beneficial task for training single-cell foundation models (Wen et al., 2023). Improving performance on imputation develops learning co-expression across genes, and (in our formulation) across cells. For evaluation, we randomly mask gene expression values at varying masking ratios by replacing them with the <MASK> token, and compare against other models. For other baselines, we apply masking and preprocessing as appropriate; CellPLM represents masked positions with 0 expression and takes several cells as input, while scGPT or scBERT only allow for gene expression values within a single cell. For MAGIC, we use the implementation from scanPy (Wolf et al., 2018) with the standard parameters.

The results of these experiments are shown in Tab. 8 for scRNA-seq data and spatial transcriptomics data respectively. In order to ensure a fair comparison with models like scGPT and CellPLM, which output continuous values, we bin the output predictions with the bin-edges that we calculate during preprocessing. CellPLM and scGPT predict only non-zero gene expression values, so we remove unexpressed genes from the evaluation. Our results demonstrate that sCT exhibits higher correlation when reconstructing partial gene sequences, and is able to better exploit information coming from multiple cells compared to CellPLM. We also evaluate the effect of using multiple cells as input, by training and testing variants of sCT with one and ten cells in Fig. 9. We observe that using 50 cells improves upon gene imputation especially in the high masking ratio regime.

In addition for spatial transcriptomics, we evaluate 'fixed' gene masking, wherein a set of genes are pre-selected, and masked for each spot across a whole study. In contrast to random masking, the model must predict the values of these unseen genes for each sample. To choose the candidate genes to be masked, we follow the experimental design in Wen et al. (2023) first suggested by Avşar & Pir (2023). This design stratifies all genes into four groups by dataset sparsity, defined as the percentage of samples wherein the gene has a zero read count. For a dataset, we take the intersection of all genes in the dataset with the set of genes observed in training, and partition them into four groups by thresholding on the sparsity ratio $s_x$ of each gene: $[s_x < 0.75, 0.75 \leq s_x < 0.90, 0.90 \leq s_x < 0.95, 0.95 \leq s_x]$. We represent this groups as Low, Medium, High and Very High respectively in

Table 3: **Zero-shot cell-typing and clustering across tissues from embeddings.** sCT is stronger at zero-shot clustering, outperforming all baselines on cell-type classification and cell-embedding clustering across four held out tissue studies, showing that model learns biologically relevant embeddings. Acc. = $k$-NN Classification accuracy, NMI = normalized mutual information, ARI = adjusted rand index.

| | Lung | | | Blood | | | Breast Cancer | | | Kidney | | |
|---|---|---|---|---|---|---|---|---|---|---|---|---|
| Model | Acc. | NMI | ARI | Acc. | NMI | ARI | Acc. | NMI | ARI | Acc. | NMI | ARI |
| sCT (zero-shot) | 0.94 | **0.67** | **0.45** | **0.82** | **0.39** | **0.19** | 0.86 | **0.39** | **0.20** | **0.99** | 0.35 | 0.06 |
| CellPLM (zero-shot) | 0.77 | 0.45 | 0.28 | 0.66 | 0.11 | 0.05 | 0.53 | 0.01 | 0.02 | 0.90 | -0.01 | 0.03 |
| Geneformer (zero-shot) | 0.90 | 0.54 | 0.32 | 0.82 | 0.36 | 0.16 | 0.68 | 0.12 | 0.07 | 0.98 | **0.36** | **0.08** |
| scGPT (zero-shot) | 0.77 | 0.29 | 0.09 | 0.71 | 0.09 | 0.03 | 0.55 | 0.02 | 0.01 | 0.89 | 0.06 | 0.01 |
| scBERT (zero-shot) | 0.66 | 0.27 | 0.09 | 0.68 | 0.09 | 0.02 | 0.45 | 0.03 | 0.01 | 0.91 | 0.17 | 0.02 |
| scVI (fitted) | **0.96** | 0.64 | 0.26 | 0.81 | 0.35 | 0.11 | 0.83 | 0.31 | 0.09 | 0.98 | 0.26 | 0.01 |
| Scanpy + Log. Reg. (trained) | 0.94 | N/A | N/A | 0.79 | N/A | N/A | 0.89 | N/A | N/A | 0.96 | N/A | N/A |
| Scanpy + $k$-NN (trained) | 0.95 | N/A | N/A | 0.81 | N/A | N/A | **0.90** | N/A | N/A | 0.96 | N/A | N/A |

our results. Next we randomly sample 25 genes from each group to add to the fixed set of genes. We select 100 total genes for each of three studies from the HEST dataset, and ask each of sCT and CellPLM to predict values for these genes, and report the MCC in Table 7 in the appendix.

**Zero-shot Whole-Cell Expression Imputation Results.** This task evaluates a model's capacity to impute the gene expression of entire cells within a spatial context. This is crucial for understanding relationships between neighboring cells in spatial transcriptomics, which has implications for realizing their therapeutic potential (Arora et al., 2023; Park & Lin, 2023). To assess this, we designed a whole-cell imputation task. Here, we mask the gene expression for one or more entire cells and challenge the model to predict these masked values based on the expression of neighboring cells. This setup mimics the array designs used in spatial transcriptomics, where gene expression is measured only at specific spots on a slide. This can lead to the undersampling or complete omission of cells, depending on their overlap with these spots. Our task simulates this effect, evaluating the model's ability to reconstruct missing gene expression profiles from the spatial context.

Table 2 presents the results for this task in both scRNA-seq and spatial transcriptomics settings. In addition to CellPLM[2], we include a per-gene heuristic inspired by $k$-nearest neighbors smoothing (Wagner et al., 2017) that leverages a $k$-NN estimator with $k = 5$. The results demonstrate that sCT effectively leverages spatial neighborhood information to predict masked cell values and reconstruct complete gene expression sequences.

**Zero-shot Cell-Typing and Clustering Results.** In this last task, we directly take cell embeddings and use them as features for both cell type prediction and clustering. For sCT , we compute each cell's embedding as the the average of multiple outputs from the transformer block, resulting in a single vector per cell. We then clustered all embedded cells using Leiden clustering (Traag et al., 2019). We assign cell-types using $k$-nearest neighbors after clustering in the embedding space. Note that this approach is training-free. We also calculate other standard clustering metrics such as normalized mutual information (NMI), and Adjusted Random Index (ARI).

We take the CellxGene data for four tissues (lung, blood, breast cancer, and kidney) and perform a 5-NN algorithm to classify the different cell types in the tissue. We report results in Tab. 3. We compare sCT 's for this task with CellPLM, Geneformer, scGPT, and scBERT. We also compare with three strong bioinformatics baselines that are trained on the test datasets: scVI (Lopez et al., 2018), and scanpy with logistic regression and $k$-nearest neighbours. We observe that sCT consistently outperforms other single-cell models on the cell type annotation task, which suggests that sCT learns representations that align with the notions of cell type in the embedding space. sCT is also comparable to the bioinformatics baselines in spite of never having trained on the test datasets. We

---

[2]The public CellPLM implementation fails to solve the whole-cell imputation task due to a scaling library-size factor that relies on some genes being unmasked. We correct for this by calculating the scaling factor over the set of cells instead of at the cell level.

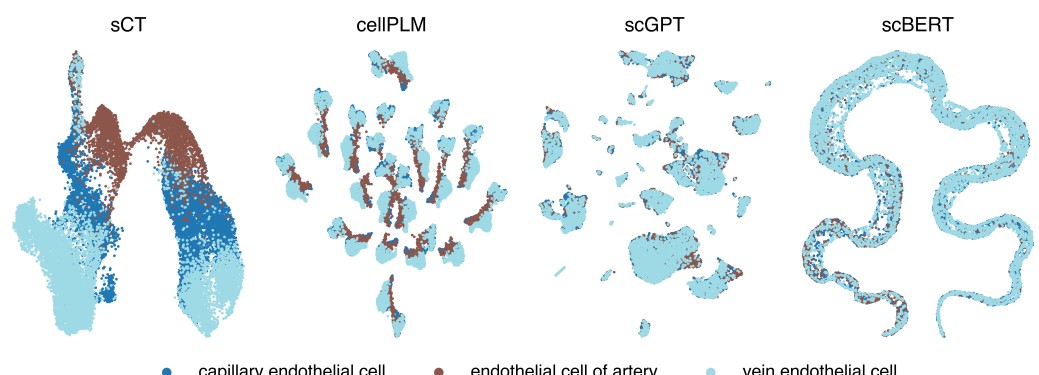

Figure 3: **sCT preserves cell types clusters**. UMAP plots for the a single-cell breast cancer study colored by cell types for sCT . We report results for sCT and three competitors. This example study was not part of the pre-training data for any of the evaluated models.

show examples of UMAP (McInnes & Healy, 2018) projections in Fig. 3. More examples can be found in Fig. 14.

**Ablation Study.**  To validate our design choices for sCT , we conducted an ablation study using the zero-shot gene imputation task. We report the average Matthews Correlation Coefficient (MCC) across all datasets as the performance indicator (see Table 4).

Table 4: **Impact of Architectural Choices on sCT Performance.** This table presents an ablation study evaluating the impact of key architectural choices on sCT 's performance. Each row modifies a specific design element while keeping others at their baseline configuration. The results below are obtained by measuring the performance on predicting gene expression levels for MCC (Matthews Correlation Coefficient) (Chicco & Jurman, 2020) is used to assess performance. More detailed plots can be found in the appendix.

| Architectural Choice | Ablation | Performance |
|---|---|---|
| All components (sCT ) | Stratified masking (1%) + Shared Layer Norm + 50 cells per sample | **0.48** |
| Masking Strategy | Stratified → Uniform masking | 0.34 |
|  | Stratified masking 1% → 5% | 0.45 |
| Layer Normalization | Shared → Standard Layer norm | 0.39 |
|  | Shared → No Layer norm | 0.40 |
| Cells per Sample | 50 cells → 1 cell per sample | 0.07 |
|  | 50 cells → 10 cells per sample | 0.44 |

First, we investigated the importance of our stratified masking strategy. Replacing it with the uniform masking strategy (15% uniform masking) commonly used to train BERT models led to a performance drop from 0.48 to 0.34. Furthermore, increasing the masking percentage of non-expressed genes from 1% to 5% reduced performance from 0.48 to 0.35, highlighting the sensitivity of the model to this parameter. Next, we examined the impact of our shared layer normalization strategy. Replacing it with a standard layer normalization (applied to all genes across all cells without considering cell symmetries) or removing layer normalization entirely resulted in performance drops to 0.39 and 0.40, respectively, from the baseline of 0.48. This underscores the importance of our shared layer normalization approach. We then analyzed the effect of stacking multiple cells in the input sequence. Using only a single cell as input significantly degraded performance (from 0.48 to 0.07). Similarly, reducing the number of stacked cells from 50 to 10 decreased performance to 0.44, demonstrating the benefit of stacking multiple cells.

Finally, we evaluated the impact of the number of bins used for gene expression discretization. While increasing the number of bins generally improved performance when estimating raw gene expression counts, it negatively impacted the model's ability to measure relative gene expression levels. This is likely due to the resulting changes in class distribution; see Sec. D.3 for a detailed analysis.

## 5    DISCUSSION AND CONCLUSIONS

This paper contributes to establishing best practices for building and evaluating single-cell models for single-cell and spatial transcriptomics data. We advocate for evaluation pipelines that assess a model's zero-shot generalization capabilities and the use of metrics robust to data imbalance. We also introduce a series of novel architectural choices for language models in this domain, culminating in the proposed sCT model. Through extensive evaluation across three tasks and 12 datasets (six single-cell and six spatial), we demonstrate that current state-of-the-art single-cell models exhibit poor generalization. In contrast, sCT shows drastically improved zero-shot generalization, outperforming even strong, trained bioinformatics baselines. Finally, we conduct a comprehensive ablation study to validate our architectural choices.

While this work represents a significant step forward, limitations remain. First, sCT does not incorporate explicit gene symbol representations, which could be valuable for regulatory network analysis. Second, we do not leverage metadata like cell type and tissue type during pretraining. We believe that incorporating such metadata would not only enhance performance but also enable the model to address complex tasks such as predicting perturbation effects and tissue-specific responses. Finally, as more experimental data becomes available, we envision expanding our zero-shot benchmark to include additional tasks like cell-to-cell communication, perturbation effect prediction, and gene regulatory network inference. We hope to explore these avenues in future work.

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

# A DATASETS

**Single-cell RNAseq data** We source scRNA-seq data from CZI CellxGene (CZI Single-Cell Biology et al., 2023) using the CxG Census API. The API allows for downloading public single-cell RNA-sequencing data across different tissues, disease types, and cell types. It also provides several filtering mechanisms that we use to ensure the quality of our dataset. We use the LTS version 2023-12-15 as our data source, which consists of approximately 62 million cells across 15,588 donors. Tab. 5 shows a tissue-wise distribution of our data. We use the same API filters as scGPT (Cui et al., 2024) to filter out duplicate cells wherever possible, to collect a total of 65 million cells. These cells are then separated by their study identifiers, followed by selecting 635 studies for training. We hold out six independent studies from different tissues: embryo, kidney, pancreas, blood, lung, and brain, for our zero-shot evaluations. The remaining studies are used for validation and hyperparameter selection. We also preprocess these studies by discarding any genes that are not well-represented in a study using the `scanpy.pp.filter_genes` with operation. Finally, we use Ensembl (Martin et al., 2022) to select only the common protein coding genes across all studies.

For each training sequence, we sample 50 cells at a time from a study, log-normalize, and then bin the gene expressions at the cellular level. These normalized sequences are then stacked into a single sequence, and input to sCT .

Table 5: **Tissuewise distribution of CellxGene Dataset.**

| Tissue | No. of normal cells (million) | No. of diseased cells (million) |
|---|---|---|
| Heart | 2.34 | 1.35 |
| Blood | 4.45 | 5.25 |
| Brain | 21.90 | 4.44 |
| Lung | 3.09 | 3.01 |
| Kidney | 0.85 | 0.73 |
| Intestine | 0.08 | 0.0 |
| Pancreas | 0.22 | 0.022 |
| Others | 14.87 | 2.39 |
| Total | 47.83 | 17.19 |

**Spatial Transcriptomics data.** We collected our spatial transcriptomics data from both the CellXGene repository (CZI Single-Cell Biology et al., 2023), as well as from the HEST 1k dataset (Jaume et al., 2024). The latter is a collection of 1108 matched spatial transcriptomics datasets (split into 552 assorted ST assays, 515 10x Visium datasets, 38 Xenium datasets, and three Visium HD datasets. Together, these data are collectively split between 535 human studies and 573 mouse studies. We focus on the human studies in Visium, downloading them via the template offered on the HuggingFace website ( `https://huggingface.co/datasets/MahmoodLab/hest`).

Tab. 6 shows a tissue-wise distribution of the spatial data. The spots in these studies are processed in the same way as our scRNA-seq data (above). We again hold out six independent studies from different tissues: lymph node, bowel, lung, kidney, brain, bowel for our zero-shot evaluations.

Each spot during training is processed in a manner similar to our scRNA-seq data (with respect to gene expression values), but we include the spatial coordinates where the spot is located on the array. In training, we also sample 50 spots at a time from each study, using euclidean distance to keep samples within local neighbourhoods (Maneewongvatana & Mount, 1999).

# B TRAINING HYPERPARAMETERS

Since there is a large data imbalance between scRNA-seq and spatial data, the training is split into two phases. First, we train sCT on scRNA-seq data using Adam optimizer with an initial learning rate of $5 \cdot 10^{-5}$ and linearly increase it to $10^{-4}$, with $10,000$ warmup steps. We then use a cosine learning rate scheduler (Loshchilov & Hutter, 2017). We train our model on approximately $10^{12}$

Table 6: **Tissuewise distribution of HEST Dataset.**

| Tissue | No. of spots |
|---|---|
| Bladder | 4086 |
| Bowel | 88688 |
| Brain | 141113 |
| Breast | 26216 |
| Cervix | 7764 |
| Eye | 3583 |
| Heart | 4247 |
| Kidney | 46986 |
| Liver | 64150 |
| Lung | 68877 |
| Lymph node | 98217 |
| Ovary | 8129 |
| Pancreas | 13001 |
| Prostate | 107985 |
| Skin | 10403 |
| Uterus | 3348 |
| Total | 696793 |

gene expressions. We use early stopping to choose the best model checkpoint based on the validation performance. Then, for spatial transcriptomics, we further train the above model on HEST data with the same learning rate schedule on approximately $1.5 \cdot 10^{11}$ gene expression levels.

## C  DETAILED RESULTS

### C.1  FIXED GENE MASKING

We report the results of the fixed gene masking experiment on spatial transcriptomics data. Following CellPLM (Wen et al., 2023), we processed each dataset and partitioned all genes into one of four groups based on their observed level of sparsity.   These genes are masked across all cells in

| **Low** | $0 \le \text{sparsity} < 0.75$ | **Medium** | $0.75 \le \text{sparsity} < 0.9$ |
|---|---|---|---|
| **High** | $0.9 \le \text{sparsity} < 0.95$ | **Very High** | $0.95 \le \text{sparsity} \le 1$ |

the test dataset. Each model must impute their values based on the remaining observed genes. sCT outperforms CellPLM, the current state-of-the-art approach on this task.

Table 7: **Fixed Gene masking (Spatial Tx) (100 genes total, 25 per sparsity group).** The table shows performance over three datasets with results over the four gene sparsity groups.

| | MCC (↑) | | | | | | | | | | | |
|---|---|---|---|---|---|---|---|---|---|---|---|---|
| | **Kidney** | | | | **Colon** | | | | **Brain** | | | |
| **Gene Sparsity Level** | Low | Medium | High | Very High | Low | Medium | High | Very High | Low | Medium | High | Very High |
| sCT | 0.15 ±0.0 | 0.14 ±0.0 | 0.08 ±0.0 | -0.01 ±0.0 | 0.51 ±0.0 | 0.40 ±0.0 | 0.20 ±0.0 | 0.18 ±0.0 | 0.36 ±0.0 | 0.24 ±0.0 | 0.05 ±0.0 | 0.03 ±0.0 |
| cellPLM | 0.24 ±0.0 | 0.10 ±0.0 | 0.05 ±0.0 | 0.06 ±0.0 | 0.49 ±0.0 | 0.33 ±0.0 | 0.05 ±0.002 | 0.00 ±0.0 | 0.00 ±0.0 | 0.16 ±0.04 | 0.06 ±0.0 | -0.01 ±0.0 |

### C.2  RESULTS PER DATASETS

We show more detailed study-wise results for the two imputation tasks. We also report plots for various masking ratios. Fig. 4, and Fig. 5 show that sCT improves upon CellPLM, and other baselines significantly.  This is especially evident for higher masking ratios, across all held-out studies.  We

repeat the same analysis for the spatial models, with sCT trained on both scRNA-seq, and spatial data. We observe the same trend as before (Fig. 6, and Fig. 7, with sCT outperforming CellPLM, and a baseline "most-common" algorithm. We also see a curious phenomenon, where masking the entire sequence still shows a non-zero MCC. We attribute this to the overall low variance across genes in scRNA-seq, and spatial datasets.

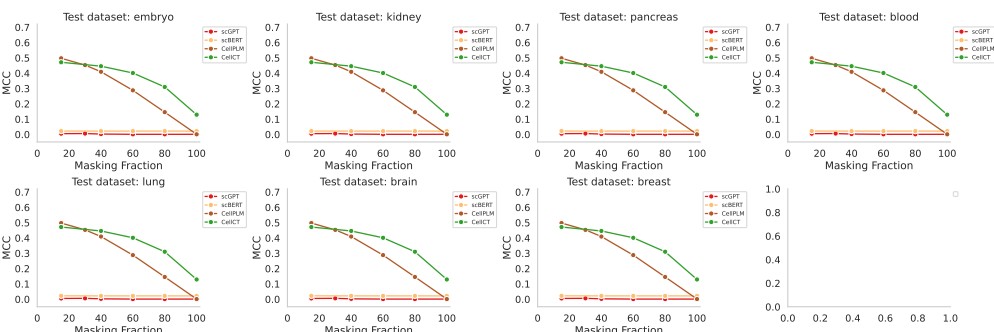

Figure 4: **Impact of the masking fraction on the MCC metric for scRNA datasets.**

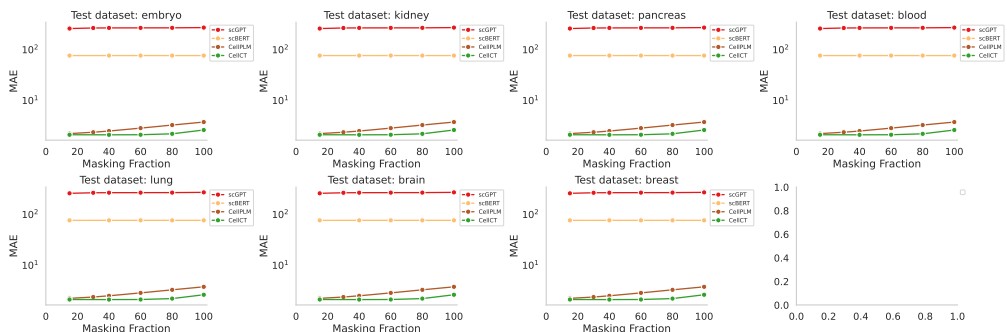

Figure 5: **Impact of the masking fraction on the MAE metric for scRNA datasets.**

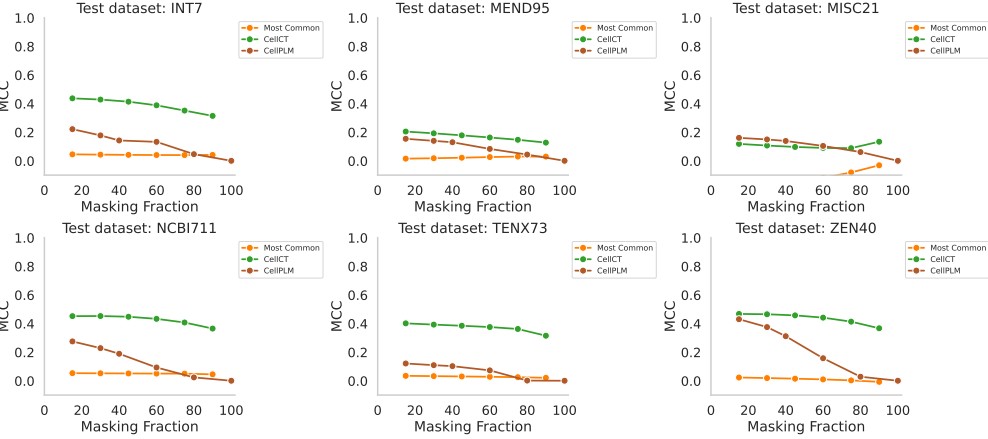

Figure 6: **Impact of the masking fraction on the MCC metric for spatial datasets.**

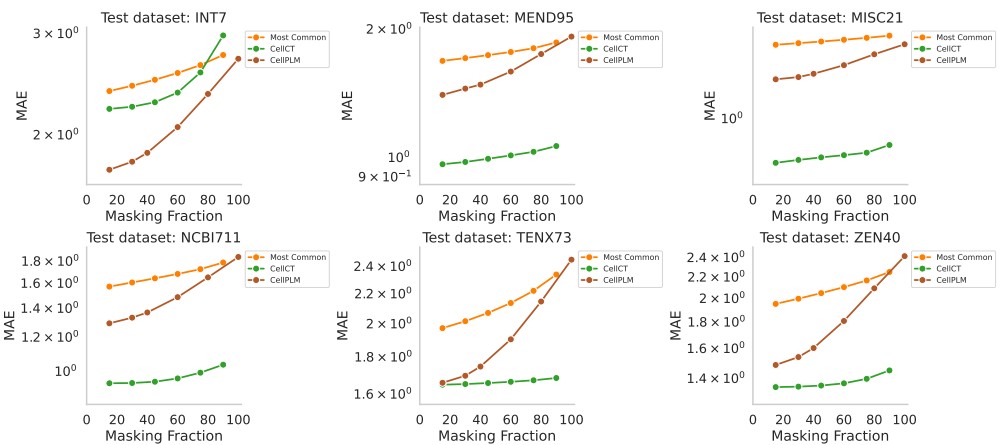

Figure 7: **Impact of the masking fraction on the MAE metric for spatial datasets.**

Table 8: **Gene imputation for different masking ratios using scRNA-seq and spatial data.** We compare sCT with literature baselines when masking a fixed fraction of genes for all cells in the input sequence during inference. Note that we only use a fixed stratified masking strategy during training. MCC = Matthews Correlation Coefficient. MAE = Mean Absolute Error. Bold font indicates that one or several algorithms are statistically better than the rest, over 5 evaluations runs. Note that no ST here refers to the base sCT pretrained on scRNA-seq data without Spatial reTraining.

| | **Masking ratios** | | | | | | | |
| | 15% | | 30% | | 40% | | 80% | |
| Model | MCC (↑) | MAE (↓) | MCC (↑) | MAE (↓) | MCC (↑) | MAE (↓) | MCC (↑) | MAE (↓) |
| **scRNA-seq** | | | | | | | | |
| sCT | **0.49** ±0.01 | **2.00** ±0.04 | **0.47** ±0.02 | **2.00** ±0.05 | **0.47** ±0.01 | **2.02** ±0.04 | **0.37** ±0.01 | **2.31** ±0.06 |
| CellPLM | **0.49** ±0.02 | 2.24 ±0.05 | 0.45 ±0.02 | 2.38 ±0.05 | 0.38 ±0.02 | 2.52 ±0.07 | 0.15 ±0.02 | 3.30 ±0.08 |
| scGPT | 0.00 ±0.001 | 260.33 ±70.05 | 0.00 ±0.001 | 266.95 ±81.53 | 0.00 ±0.002 | 267.88 ±73.48 | 0.00 ±0.002 | 268.46 ±92.31 |
| scBERT | 0.04 ±0.01 | 76.59 ±14.32 | 0.04 ±0.002 | 76.64 ±12.86 | 0.03 ±0.01 | 76.70 ±13.24 | 0.02 ±0.01 | 76.98 ±11.29 |
| **Spatial Transcriptomics (ST)** | | | | | | | | |
| sCT (only scRNA-seq) | 0.05 ±0.01 | 1.40 ±0.05 | 0.05 ±0.01 | 1.41 ±0.05 | 0.04 ±0.01 | 1.42 ±0.05 | 0.03 ±0.01 | 1.51 ±0.05 |
| sCT (scRNA-seq + ST) | **0.35** ±0.03 | **1.31** ±0.05 | **0.34** ±0.02 | **1.32** ±0.05 | **0.33** ±0.02 | **1.33** ±0.05 | **0.28** ±0.02 | **1.45** ±0.06 |
| CellPLM | 0.23±0.02 | 1.48±0.05 | 0.20 ±0.02 | 1.52±0.05 | 0.17±0.01 | 1.71 ±0.06 | 0.03 ±0.01 | 2.02 ±0.07 |

Table 9: **Cell imputation for different numbers of masked cells using both scRNA-seq and spatial transcriptomics data.** We compare sCT with CellPLM when masking all gene expression values (100% masking) for a given number of cells during inference. Predictions are based only on scRNA-seq data. MCC = Matthews Correlation Coefficient (higher is better). MAE = Mean Absolute Error (lower is better).

| | **Number of masked cells** | | | | | | | |
| | 1 | | 10 | | 20 | | 40 | |
| Model | MCC (↑) | MAE (↓) | MCC (↑) | MAE (↓) | MCC (↑) | MAE (↓) | MCC (↑) | MAE (↓) |
| **scRNA-seq** | | | | | | | | |
| sCT | **0.70** ±0.04 | **1.64** ±0.06 | **0.71** ±0.04 | **1.85** ±0.06 | **0.63** ±0.03 | **2.07** ±0.06 | **0.43** ±0.02 | **2.53** ±0.08 |
| CellPLM | 0.00 ±0.01 | 3.78 ±0.08 | 0.00 ±0.01 | 3.78 ±0.08 | 0.00 ±0.01 | 3.78 ±0.08 | 0.00 ±0.01 | 3.78 ±0.08 |
| CellPLM[+] | 0.09 ±0.02 | 3.54 ±0.07 | 0.06 ±0.02 | 3.64 ±0.07 | 0.02 ±0.01 | 3.57 ±0.07 | 0.02 ±0.01 | 3.55 ±0.07 |
| $k$-NN smoothing | 0.11 ±0.02 | 2.80 ±0.06 | 0.05 ±0.01 | 3.03 ±0.06 | 0.07 ±0.02 | 3.21 ±0.07 | 0.06 ±0.01 | 3.51 ±0.09 |
| **Spatial Transcriptomics (ST)** | | | | | | | | |
| sCT | **0.57** ±0.03 | **1.25** ±0.05 | **0.54** ±0.03 | **1.26** ±0.05 | **0.46** ±0.03 | **1.29** ±0.05 | **0.32** ±0.02 | **1.36** ±0.05 |
| CellPLM | 0.00 ±0.00 | 2.13 ±0.06 | 0.00 ±0.00 | 2.15 ±0.06 | 0.0 ±0.00 | 2.17 ±0.06 | 0.00 ±0.00 | 2.16 ±0.06 |
| CellPLM[+] | 0.13 ±0.02 | 2.12 ±0.06 | 0.13 ±0.02 | 2.10 ±0.06 | 0.13 ±0.02 | 2.05 ±0.06 | 0.12 ±0.02 | 1.84 ±0.05 |
| $k$-NN smoothing | 0.00 ±0.01 | 1.87 ±0.05 | 0.01 ±0.01 | 1.92 ±0.05 | 0.01 ±0.00 | 1.98 ±0.05 | 0.03 ±0.01 | 2.18 ±0.06 |

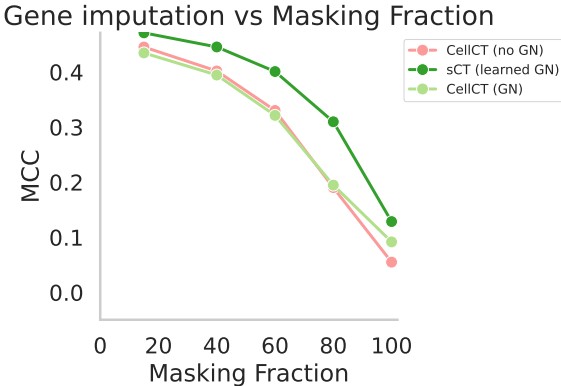

Figure 8: **Impact of learned gene normalization**

## D COMPLEMENTARY RESULTS ON THE ABLATION STUDY

### D.1 IMPACT OF LEARNED GENE NORMALIZATION

In Fig. 8, we show the effect of using our shared layer norm layer. We compare the base sCT (trained with shared layer norm), with two baselines, one being trained with a gene normalization layer without learnable parameters, and another with no gene normalization at all. We see that shared layer norm improves performance significantly over both baselines. However, we do observe this curious phenomenon where a model with no normalization, and one with, tend to perform almost the same, leading us to conjecture that shared layer norm is perhaps learning gene-level statistics across our training data.

We also show tissue-wise distribution for a comparison between a model trained with the shared layer norm, and one otherwise. We see a consistent improvement on held-out test performance across all tissues.

Table 10: **Shared layer norm is a superior to standard layer norms**. We compare the Matthews Correlation Coefficient (MCC) for the same sCT configuration using the proposed Learned Gene Norm and the standard Layer Norm across different tissues. A 15% masking rate is used in all cases.

| Tissue | Blood | Brain | Breast | Embryo | Kidney | Lung | Pancreas | Average |
|---|---|---|---|---|---|---|---|---|
| **Shared layer norm** | **0.43** | **0.55** | **0.46** | **0.27** | **0.44** | **0.49** | **0.27** | **0.42** |
| **Layer Norm** | 0.36 | 0.46 | 0.37 | 0.20 | 0.36 | 0.40 | 0.20 | 0.34 |

### D.2 IMPACT OF THE NUMBER OF CELLS PER SAMPLE

Expanding upon results in Tab. 4, we show that ingesting gene expression sequences for multiple cells improves gene imputation performance. For this experiment, we train two additional variants of sCT , with one, and ten cells as input during training. We then repeat our gene imputation evaluations for the these two models and compare with the base model trained with 50 cells, across a variety of masking fractions. We observe in Fig. 9 that adding even 10 cells significantly improves performance across all masking fractions over a single cell as input. However, the 50-cell variant outperforms both, especially as more genes are randomly masked. This provides strong evidence for the design of our framework.

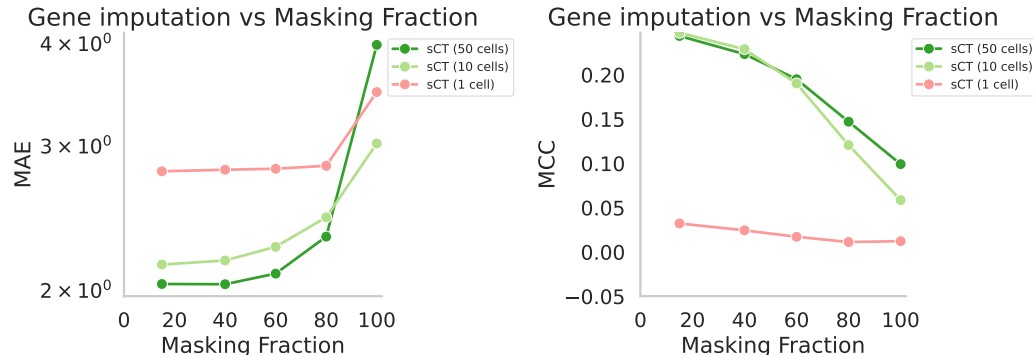

Figure 9: **Impact of the number of cells per sample on both the MCC and the MAE metrics for scRNA datasets.**

### D.3 IMPACT OF THE NUMBER OF BINS

A core part of sCT is that we bin gene expressions at a cellular level, and treat them as tokens. Particularly, we choose five bins to represent unexpressed, low expression, moderate expression, high-moderate expression, and high expression levels for every cell. We present an ablation of this hyperparameter by also trying a larger number of bins (15 bins). This represents a finer-grained categorization of gene expressions in a cell. We measure mean-squared log error (MSLE), and mean absolute error (MAE) for the two models. Fig. 10 shows that we can improve our estimations of continuous-valued gene expression counts by using more bins.

However, given the sparsity of the gene expression sequence of a cell, uniformly binning gene expressions often leads to several gene expression levels being under-represented in our training data. We find that this affects our performance on our predicting gene expression levels when we consider the MCC metric, leading to noisy estimates (see Fig. 10(c)). We therefore discretize gene expressions into five bins.

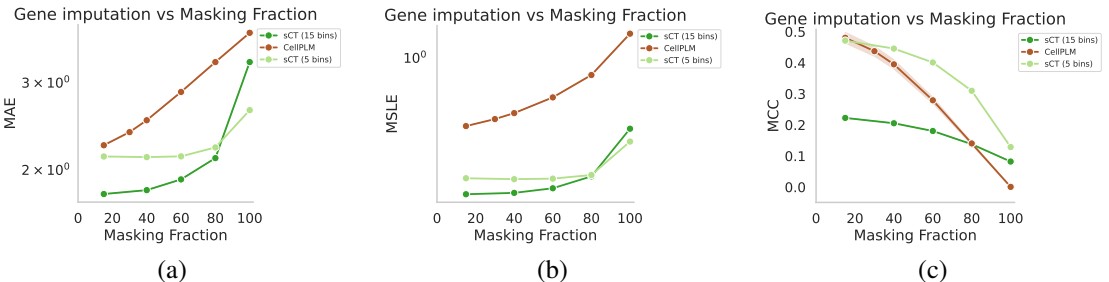

Figure 10: **Impact of number of bins.**

### D.4 IMPACT OF STRATIFYING MASKING

We also analyze the effect of our stratified masking approach in Fig. 11 by comparing sCT trained with two separate masking ratio schemes: (1) 1% zeros masked, 15% non-zeros masked, and (2) 5% zeros masked, 15 % non-zeros masked. In the first case, we arrive at the masking ratios by analyzing the training dataset and calculating the relative distribution of unexpressed versus expressed genes. We find that this is $\approx 1 : 15$ for our training split. We trained the second model to study if masking more unexpressed genes is really helpful, as the relative information contained in unexpressed gene expressions is fairly low. We see that using a stratified masking strategy helps when estimating discrete gene expression levels, especially at higher masking ratios. However, the performance on estimating raw gene expression levels is relative unaffected at higher masking ratios. We note that both the models outperform CellPLM (Wen et al., 2023).

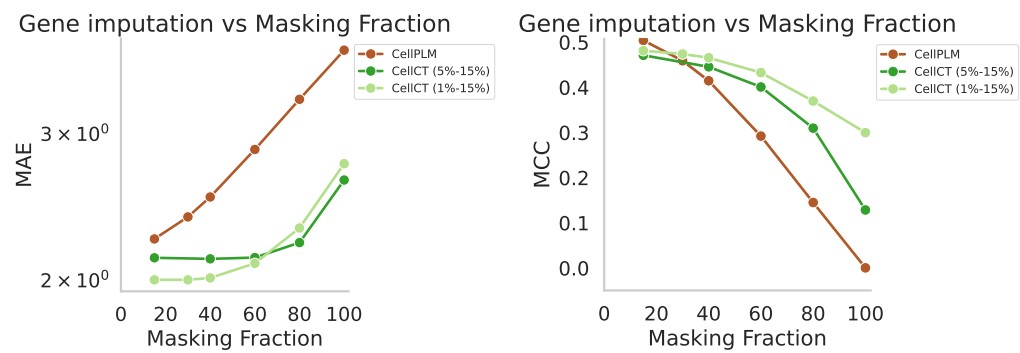

Figure 11: **Impact of stratified masking.**

## E    COMPLEMENTARY RESULTS FOR CLUSTERING

Find below additional comparisons of sCT clustering capabilities versus competitors methods on the following additional studies on kidney,blood and lung tissues. Clusters are colored by cell types. Please note that as for study on breast tissue showed in the main core of the paper, none of these studies were present in any of the evaluated methods to assess zero-shot capabilities.

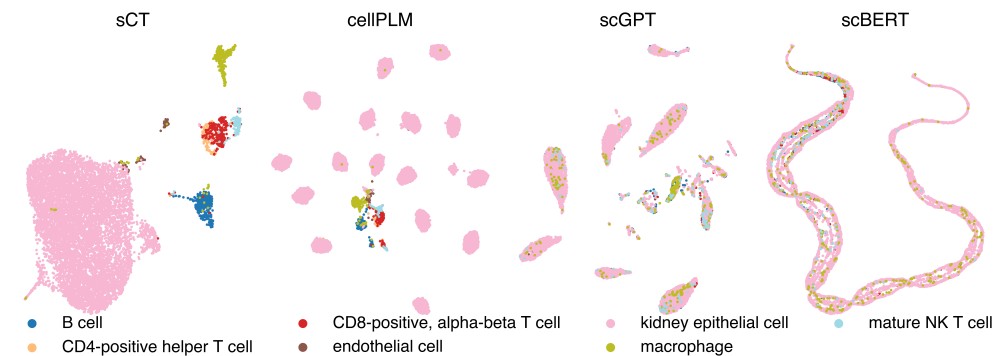

Figure 12: Additional UMAP plots for the Kidney study colored by cell types for sCT and three competitors.

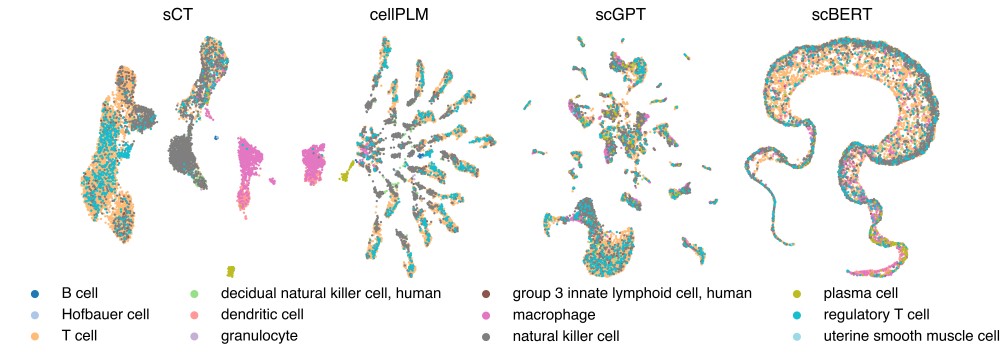

Figure 13: Additional UMAP plots for the Blood study colored by cell types for sCT and three competitors.

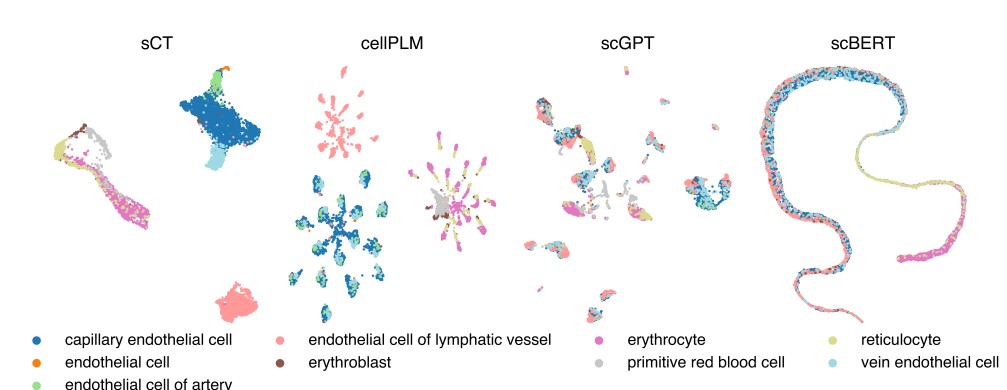

Figure 14: Additional UMAP plots for the Lung study colored by cell types for sCT and three competitors.

