# OpenReview forum: "A long range foundation model for zero-shot predictions in single-cell and spatial transcriptomics data"
_ICLR.cc/2025/Conference — Submitted to ICLR 2025_

### Official Review · Reviewer_ezip · 2024-10-27

**Soundness:** 1
**Presentation:** 2
**Contribution:** 1
**Rating:** 3
**Confidence:** 4

**Summary:**

Here the authors present sCellTransformer (sCT), a transformer model for analyzing single-cell RNA-seq and spatial transcriptomics data. sCT is distinguished from previously proposed "single-cell foundation models" through its use of a convolutional transformer architecture that scales to handle all ~20,000 protein coding genes per cell. The authors apply their model to a set of imputation/clustering tasks.

**Strengths:**

* **Architectural choices**: The authors incorporate new architectural ideas (e.g. from Linder et al. 2023) to enable the use of all ~20,000 protein coding genes with single-cell transformer methods.
* **Incorporation of spatial information**: To my knowledge, sCT is the first single-cell transformer that specifically attempts to incorporate spatial information via spatial positioning tokens.

**Weaknesses:**

After reading the paper I have numerous major concerns that prevent me from recommending acceptance. In short, the presented results suffer from a number of common issues in recent "single-cell foundation model" papers (e.g. lack of rigorous benchmarking against current best practice methods), and it's not clear to me that sCT represent a major advance for the analysis of single-cell data. Details below:

* **Missing comparisons with non-"foundation model" baselines**: A number of recent benchmarking studies (e,g, [1, 2, 3] among others) have found that recently published transformer-based single-cell foundation models perform no better than significantly simpler (e.g. linear) baseline models at a variety of downstream tasks. However, the authors' manuscript only compares against other transformer models, thus making it impossible to assess whether the authors' method indeed represents a meaningful advancement. For for the batch effect/cell type classification tasks, how do simpler baselines (e.g. logistic regression, scANVI, etc.) compare?
* **Unclear real-world utility**: Based on the provided experimental results, it's not clear to me that the authors' method provides additional utility beyond current standard scRNA-seq analysis workflows. For example, given that (as mentioned by the authors) <6% of protein-coding genes are expressed in an individual cell on average, I'm not sure why the results presented in Table 2 are meaningful, as most of the time all the masked values will have zero measured expression. Similarly, I'm not clear on what real-world need the authors are trying to address in the "whole-cell imputation task". The authors mention that measurements from some individual cells can be missed due to misalignment with spot boundaries, but it's not clear to me why this phenomenon would lead to an entire spot dropping out. The bottom line is: what **new** analyses are enabled by the authors' approach? As-is I'm sure what sCT brings to the table compared to previous approaches, beyond (maybe) better imputation performance.
* **Preprocessing/evaluation details**: For preprocessing the authors mention only applying a log transform to raw expression values and then tokenizing via a uniform binning strategy. How do the authors handle differences in sequencing depth/library size between cells? From the description provided in the manuscript I don't see which preprocessing steps (if any) handle this issue, which could confound the authors' tokenization scheme. On a related note, the performance of some baseline methods is bad enough that I'm concerned the reported results may be an artifact of the authors tokenization/evaluation setup. For example the results for scGPT in Table 2 show mean absolute error values of 260, which seems suspicious to me given that the data has been log-transformed/binned. Are the scGPT-outputted values on a significantly different scale than the bin values calculated for sCT during preprocessing? Or as another sanity check, what do the corresponding errors look like for simple baseline of prediction zero expression or the mean expression for that gene?

[1]: Kedzierska et al. "Assessing the limits of zero-shot foundation models in single-cell biology"
[2]: Kernfeld et al. "A systematic comparison of computational methods for expression forecasting"
[3]: Ahlmann-Eltze et al. "Deep learning-based predictions of gene perturbation effects do not yet outperform simple linear methods"

**Questions:**

See "Weaknesses".

---

> ### Author Response · Authors · 2024-11-27
> **Response to reviewer ezip (1 / 2)**
>
> We thank the reviewer for their helpful criticism.  We address each point below:
>
> > After reading the paper I have numerous major concerns that prevent me from recommending acceptance. In short, the presented results suffer from a number of common issues in recent "single-cell foundation model" papers (e.g. lack of rigorous benchmarking against current best practice methods), and it's not clear to me that sCT represent a major advance for the analysis of single-cell data. Details below:
>
> Thank you for your feedback. We understand your concerns regarding current single-cell foundation models. We would like to clarify that our primary goal in this paper was not to achieve a major breakthrough in single-cell data analysis. We believe such an expectation would be overly ambitious for a machine learning paper at a conference like ICLR.
>
> Instead, we positioned our work within the literature of self-supervised models trained on single-cell and spatial transcriptomics data. Our objectives were twofold:
>
> 1. To evaluate the zero-shot capabilities of such models on tasks aligned with their pretraining objectives but applied to new data domains.
> 2. Motivated by the observation that state-of-the-art models like scGPT struggle in these scenarios, to propose machine learning advancements that address these limitations.
>
> We believe this contribution is well-suited for a conference like ICLR and helps establish better practices for researchers developing such models.
>
> > Missing comparisons with non-"foundation model" baselines: A number of recent benchmarking studies (e,g, [1, 2, 3] among others) have found that recently published transformer-based single-cell foundation models perform no better than significantly simpler (e.g. linear) baseline models at a variety of downstream tasks. However, the authors' manuscript only compares against other transformer models, thus making it impossible to assess whether the authors' method indeed represents a meaningful advancement. For for the batch effect/cell type classification tasks, how do simpler baselines (e.g. logistic regression, scANVI, etc.) compare?
>
> Thank you for raising these concerns. We have now incorporated additional bioinformatics baselines for gene imputation (MAGIC, Table 1) and cell typing (scVI, Scanpy + logistic regression, and k-nearest neighbors in Table 3). The results are reproduced in our reply to all reviewers above, as well as in the revised manuscript.
>
> It is important to highlight that each of these baselines requires either training (sometimes referred to as "fitting") on the target data or the use of additional metadata. In contrast, we evaluate sCT in a zero-shot setting, ensuring that the evaluation data was not present in the sCT training data. Despite this significant constraint, sCT matches or outperforms these baselines. We also emphasize that comparable single-cell foundation models underperform these baselines in zero-shot scenarios.
>
> We hope these results clarify our contributions and demonstrate the effectiveness of our approach.
>
> > Unclear real-world utility: Based on the provided experimental results, it's not clear to me that the authors' method provides additional utility beyond current standard scRNA-seq analysis workflows. For example, given that (as mentioned by the authors) <6% of protein-coding genes are expressed in an individual cell on average, I'm not sure why the results presented in Table 2 are meaningful, as most of the time all the masked values will have zero measured expression
>
> We agree with the reviewer observation that, on average, less than 6% of protein-coding genes are expressed in an individual cell. This sparsity was a key motivator for many of our design choices and evaluation metrics. Specifically, it informed our decision to exclusively mask non-zero expressed genes in all cell imputation experiments. Thus, when we refer to masking $k\\%$ of the genes, this signifies masking $k\\%$ of the non-zero protein-coding genes. We have further emphasized this point in the manuscript.

---

> > ### Author Response · Authors · 2024-11-27
> > **Response to reviewer ezip (2/2)**
> >
> > > Similarly, I'm not clear on what real-world need the authors are trying to address in the "whole-cell imputation task". The authors mention that measurements from some individual cells can be missed due to misalignment with spot boundaries, but it's not clear to me why this phenomenon would lead to an entire spot dropping out. The bottom line is: what new analyses are enabled by the authors' approach? As-is I'm sure what sCT brings to the table compared to previous approaches, beyond (maybe) better imputation performance.
> >
> > Thank you for your comment. Our motivation for the whole-cell imputation task extends beyond addressing technical errors and dropout in spatial transcriptomics. This task also serves to evaluate whether sCT and other foundation models learn inter- and intra-cellular co-expression patterns.
> >
> > Moreover, we propose that this approach could be used to enhance the resolution of spatial data by interpolating between spots. We simulate this by systematically dropping out spots in our data and reconstructing them from their neighbors. Our results demonstrate that sCT significantly outperforms other baselines in this task. We have clarified these points in the manuscript.
> >
> >
> > > Preprocessing/evaluation details: For preprocessing the authors mention only applying a log transform to raw expression values and then tokenizing via a uniform binning strategy. How do the authors handle differences in sequencing depth/library size between cells? From the description provided in the manuscript I don't see which preprocessing steps (if any) handle this issue, which could confound the authors' tokenization scheme. On a related note, the performance of some baseline methods is bad enough that I'm concerned the reported results may be an artifact of the authors tokenization/evaluation setup. For example the results for scGPT in Table 2 show mean absolute error values of 260, which seems suspicious to me given that the data has been log-transformed/binned. Are the scGPT-outputted values on a significantly different scale than the bin values calculated for sCT during preprocessing? Or as another sanity check, what do the corresponding errors look like for simple baseline of prediction zero expression or the mean expression for that gene?
> >
> > We thank the reviewer for raising these important points. We have clarified our preprocessing approach in the manuscript, specifically in Section 3, and address the reviewer's concerns below:
> >
> > **Handling Differences in Sequencing Depth/Library Size:**
> > Our discrete binning approach operates at the cellular level, allowing the model to effectively disregard variations introduced by sequencing depth between cells. During inference, when predicting raw counts, we leverage bin edges, which inherently encode sequencing depth or library size information and capture inter-cell variability. Although this information is not explicitly encoded during training, our binning strategy, combined with post-processing (unbinning with cellular sequencing depths), leads to improved predictive performance compared to directly predicting raw counts.
> >
> > **Disparities in Results:**
> > We have updated the manuscript (Section 4) to clarify that the reported mean absolute error (MAE) values are computed on raw counts, not log-transformed data. This ensures fair comparisons across all models. Additionally, we confirm that the same bin edges are used across models for MCC metric calculations. Regarding scGPT’s performance,  our findings align with recent literature, which also reports that scGPT strongly underperforms on zero-shot gene expression prediction tasks (see Section 3.3 in [1]), supporting the observed results.
> >
> > We hope these clarifications address the reviewer’s concerns.
> >
> > [1]: Kedzierska et al. "Assessing the limits of zero-shot foundation models in single-cell biology"

---

> > > ### Comment · Reviewer_ezip · 2024-12-02
> > >
> > > Dear authors,
> > >
> > > Thank you for your response and for the additional experimental results. I am choosing to keep my original score, as my concerns regarding benchmarking and the downstream utility of the authors' method remain unresolved.
> > >
> > > Indeed, the new results appear to validate what previous works have already found: simple baseline workflows (e.g. the basic scanpy workflow on cell type classification) can provide comparable performance on real-world scRNA-seq tasks of interest at a fraction of the computational cost of transformer-based single-cell models. Moreover, on tasks where the authors' method does outperform baselines, it's unclear to me how robust these results are, as it appears that all presented results correspond to pretraining with a single random seed; previous work [1] has found that evaluation results for single-cell transformer models can vary substantially for different training seeds.
> > >
> > > While it is interesting that the authors' results are under a zero-shot setting, given the substantially higher cost of running these models even after pretraining (e.g. in terms of GPU memory), it's not clear to me that the practical benefits of the zero-shot setting outweigh the cost of applying a less computationally intensive pipeline on a new dataset. Thus, I believe the submitted work does not represent a meaningful enough advancement to be presented at ICLR.
> > >
> > > [1]: Boiarsky et al., "A Deep Dive into Single-Cell RNA Sequencing Foundation Models"

---

> > > > ### Author Response · Authors · 2024-12-02
> > > > **Reply to reviewer ezip**
> > > >
> > > > We sincerely thank the reviewer for their feedback and for taking the time to review our additional results.
> > > >
> > > > Regarding the concerns about baselines and pretraining costs, we agree that our experiments support the conclusion that simple workflows can provide competitive results for some tasks. However, we emphasize that our results indicate sCT achieves comparable performance in **zero-shot settings**, requiring no task-specific training. This eliminates the need for retraining or fitting on new datasets, thereby amortizing the pretraining cost. Moreover, the relatively small size of our model (~80M parameters) allows for cost-effective inference on CPUs or smaller GPUs, making it as accessible as traditional methods.
> > > >
> > > > On robustness, we respectfully disagree with the comparison to prior work in [1]. That study evaluated the sensitivity of **fine-tuned models**, while our results focus on **zero-shot performance**, which inherently reduces variability due to hyperparameter tuning. To ensure reproducibility, we provide all hyperparameter settings and detailed protocols for generating the reported results. We believe this demonstrates the reliability of our approach.
> > > >
> > > > Thank you again for your constructive comments. We hope these clarifications address your concerns and shed light on the potential of sCT for advancing single-cell transcriptomics research.

---

### Official Review · Reviewer_PjDS · 2024-10-30

**Soundness:** 2
**Presentation:** 2
**Contribution:** 2
**Rating:** 6
**Confidence:** 5

**Summary:**

In this study, the authors pretrained a convolutional-transformer combined model on both single cell and spatial transcriptomics data. Quantitative comparison with previous models on the imputation and clustering tasks showed the advance of the current model. Several ablation experiments motivate specific choices in the model.

**Strengths:**

The authors proposed several architecture-level improvements including convolutional layer and learned gene normalization. The ablation experiments on certain aspects are well-designed and provide insights into the model design. The author also extended the imputation task to the whole cell level.

**Weaknesses:**

First of all, current downstream tasks can not support the claim of a foundation model; it is more like an imputation model. A foundation model should deal with various tasks. Also, some concepts like "a first principles formulation of the problem" are not clearly explained.

While new convolutional architecture is interesting, the authors did not verify its effectiveness by comparing it with simple MLP, and also did not clearly explain the reason behind the design. The authors also did not fully compare the different gene expression discretization methods, making it hard to show its novelty. At least the authors should refer to the previous works on the discretization methods (https://openreview.net/pdf?id=gdwcoBCMVi). Also for the imputation task, some important baseline methods are missing, such as gimVI (https://arxiv.org/abs/1905.02269), scFoundation (https://www.nature.com/articles/s41592-024-02305-7), which are the foundation and also domain specific models claimed to deal with the missing data.

Some critical details are missing for the spatial imputation experiment design. For example, how did the author use other compared methods, did these models fine-tuned the same training data? Also, the authors did not provide detailed information on held-out data in HEST, as some of the fovs are actually replicates of the training data, which may lead to the risk of data leakage.

**Questions:**

1. Could the author further explain the concept of "a first principles formulation of the problem"? or it is better to remove this claim in the abstract.

2. What is the rationale behind the design of the new convolutional architecture? As gene expression data is nonsequential, how can the author define the "local" and "global" patterns in the data? Also, could you compare it with simple MLP to show its effectiveness?

3. Could the author further explain the difference between learned gene normalization and the original layer normalization? What is the meaning of "subsequence"? How does the learned gene normalization help the model?

4. Could the author justify the choice of the discretization method? How did you decide the bing edges? How does it compare with the previous works on discretization methods?

5. Since the expression distribution is quite different between the two data types, how can the author justify the contribution of single cell data to the spatial imputation tasks? I would like to see the performance of sCT pretrained on the spatial data only.

6. For the imputation benchmark, could the author compare with the gimVI and scFoundation, which are models that can deal with the missing data?

7. Still in the imputation tasks, did the author fine-tune the compared models on the same training data? It should be clarified in the paper.

8. Could the author provide the details of the held-out data in HEST? I suggest the author leave out FOV from new studies to avoid the risk of data leakage.

9. Since the cell type information is not used in the training process, how can the author assign cell types based on the training-free cell embeddings?

10. Some incomplete sentences in the paper, such as "with a decay rate of ...." could be revised.

---

> ### Author Response · Authors · 2024-11-27
> **Response to reviewer PjDs (1 / 3)**
>
> We thank the reviewer for their helpful critique of our work.  We address each comment specifically below:
>
> > First of all, current downstream tasks can not support the claim of a foundation model; it is more like an imputation model. A foundation model should deal with various tasks. Also, some concepts like "a first principles formulation of the problem" are not clearly explained.
>
> Thank you for this insightful comment. We agree and have toned down the use of the term "foundation model" in the paper. Our goal was to investigate the recent generation of self-supervised models trained on single-cell and spatial transcriptomics data. Inspired by work in other fields, such as proteomics and genomics, we sought to understand how well these models generalize in a zero-shot setting to new data domains. We believe this capacity for generalization is essential for the practical utility of such models.
>
> Our analysis revealed that state-of-the-art models like scGPT perform poorly when evaluated on new data domains, even for simple tasks like gene imputation, which aligns with their pretraining objective. This motivated us to further benchmark models to assess generalization capabilities and propose architectural modifications to address these limitations.
>
> Thank you also for your comment regarding the phrase "first principles." We have removed this phrase and clarified our intended meaning. Our aim was to design an approach that minimizes assumptions when processing data to avoid introducing biases. Specifically, we sought to process the expression of all coding genes for all cells in the input without resorting to subset selection and to minimize data preprocessing. We have clarified this in Section 3 and the introduction.
>
> > While new convolutional architecture is interesting, the authors did not verify its effectiveness by comparing it with simple MLP, and also did not clearly explain the reason behind the design.
>
> Thank you for raising this point. Unfortunately, using a standard multi-layer perceptron (MLP) in this context is computationally infeasible.
>
> Our input consists of $k \times n$ gene expression tokens, where $k$ represents the number of cells $(50)$ and n represents the number of coding genes $(20,000)$. This results in $10^6$ tokens. As noted in the manuscript, processing this input directly with a Transformer model is impractical due to the quadratic scaling of self-attention with respect to sequence length.
> To address this, we employ a convolutional tower to reduce the input sequence length from $10^6$ tokens to $4096$ tokens, a size manageable for processing on a single high-performance GPU.
>
> Achieving this reduction with an MLP would require transforming an input of shape `(10^6, token_embed_dim)` to `(4096, transformer_embed_dim)`. Even a single-layer MLP would necessitate `4 x 10^9 x token_embed_dim x transformer_embed_dim` parameters. Even with minimal embedding dimensions of 1, this would result in `4 x 10^9` parameters in the MLP alone, which would be roughly 50 times more than the current total number of parameters (70 million) in the model. We have clarified this constraint in the manuscript.
>
> Conversely, convolutional layers provide a mechanism for gradually reducing the sequence length while increasing the embedding dimension from `token_embed_dim` to `transformer_embed_dim`. This is achieved with a manageable number of parameters due to weight sharing within the convolutions. Transpose convolutions then enable us to reverse this process, expanding the sequence length back to its original size while reducing the embedding dimension for gene expression reconstruction. This architecture also allows for skip connections between the downsampling and upsampling layers.
>
> While we acknowledge that using convolutions for unordered data like ours might seem counterintuitive, our primary motivation was dimensionality reduction. We were pleased to find that this architecture not only effectively downsamples and upsamples the sequence with a controlled number of parameters but also demonstrates strong empirical performance. We believe this approach can serve as valuable guidance for others developing similar models. We have clarified the rationale behind this architectural choice in the manuscript.

---

> > ### Author Response · Authors · 2024-11-27
> > **Response to reviewer PjDs (2/3)**
> >
> > > The authors also did not fully compare the different gene expression discretization methods, making it hard to show its novelty- At  least the authors should refer to the previous works on the discretization methods (https://openreview.net/pdf?id=gdwcoBCMVi)
> >
> > Thank you for bringing this to our attention. We have now included the reference you provided in the manuscript to better contextualize our work.
> >
> > We would like to clarify that our gene expression discretization method is not novel and has been employed in previous work, such as scGPT and scBERT. Our key contributions lie in how we leverage this discretized data within our model architecture:
> >
> > **Embeddings:** We treat each data bin as a distinct token and learn token embeddings, adhering to best practices in Transformer and language modeling.
> >
> > **Prediction and loss function:** We predict probability distributions over tokens and utilize a cross-entropy loss, which aligns with established best practices, rather than predicting continuous values with a mean squared error loss.
> >
> > **Classification Evaluation Metrics:** We introduce the use of classification metrics like Matthews Correlation Coefficient (MCC) to robustly assess performance, especially given the imbalanced nature of the data. We believe this is an important practice that should be widely adopted in the field.
> >
> > We have clarified these points in the manuscript. Thank you for your valuable feedback!
> >
> > > Also for the imputation task, some important baseline methods are missing, such as gimVI (https://arxiv.org/abs/1905.02269), scFoundation (https://www.nature.com/articles/s41592-024-02305-7), which are the foundation and also domain specific models claimed to deal with the missing data
> >
> > Thank you for providing these references. We have included them in the paper. Unfortunately, due to the timeframe of the rebuttal period, we were unable to incorporate scFoundation into our evaluation, but we acknowledge its relevance as a baseline for future work.
> >
> > Regarding gimVI, our understanding is that it requires matched tissue samples for both scRNA-seq and spatial transcriptomics data. This requirement was not met by our datasets. Therefore, to provide an additional bioinformatics baseline, we included MAGIC, a graph-based imputation method, in our benchmark.
> >
> > > Some critical details are missing for the spatial imputation experiment design. For example, how did the author use other compared methods, did these models fine-tuned the same training data? Also, the authors did not provide detailed information on held-out data in HEST, as some of the fovs are actually replicates of the training data, which may lead to the risk of data leakage
> >
> > Thank you for raising these important points. To ensure a fair comparison in the spatial imputation experiments, we fine-tuned CellPLM on the same HEST training dataset used to train sCT.
> >
> > Regarding the held-out dataset, we carefully selected one field of view (FOV) from each data source in HEST. We manually examined the metadata to confirm that the six FOVs used for evaluation are unique and not present in the training data, thus preventing data leakage.
> >
> > We clarified these points in the experimental section of the revised manuscript.

---

> > > ### Author Response · Authors · 2024-11-27
> > > **Response to reviewer PjDs (3/3)**
> > >
> > > We address the remaining questions below:
> > >
> > > > Since the expression distribution is quite different between the two data types, how can the author justify the contribution of single cell data to the spatial imputation tasks? I would like to see the performance of sCT pretrained on the spatial data only.
> > >
> > > Thank you for this insightful comment. We have initiated training sCT on spatial data only. However, the training process has not yet completed at the time of writing. We will update this response with the results once they are available, which we anticipate will be by the end of the week.
> > >
> > > > Since the cell type information is not used in the training process, how can the author assign cell types based on the training-free cell embeddings?
> > >
> > > Thank you for raising this point. The cell type labels are available in our test data but are not provided to sCT during training or evaluation. At evaluation time, we perform inference with sCT on each cell's gene expression profile and extract the corresponding cell embeddings. We then apply unsupervised clustering algorithms in the embedding space and assess the concordance of these clusters with the held-out cell type labels. This evaluation methodology follows that of CellPLM. We have clarified these points in the manuscript.
> > >
> > > > Some incomplete sentences in the paper, such as "with a decay rate of ...." could be revised.
> > >
> > > Thank you for identifying these typos. We have corrected them and further edited the manuscript to improve readability and clarity.

---

> ### Author Response · Authors · 2024-12-02
> **Summarized response to reviewer PjDS**
>
> Dear reviewer PjDS,
>
> Thank you for constructive feedback on our paper. We appreciate your time and consideration towards reviewing our work. In response to your comments, we summarize our rebuttal below and highlight the changes in our manuscript.
>
> **Terminology clarifications:**  We replaced "foundation model" with more precise descriptions and clarified our goal of improving generalization in self-supervised models for single-cell RNA-seq data. We also modified section 3 in our manuscript to clarify our contributions.
>
> **Clarifications on Architecture:** We justified the use of convolutional layers over MLPs for dimensionality reduction due to computational constraints. We also highlighted the efficiency and empirical performance of the convolutional design.
>
> **Imputation and Baseline Methods:** We have added references to gimVI and scFoundation.  We have also added additional baselines for gene imputation and cell-typing studies.
>
> **Experimental Design and Data Leakage:** We confirmed no data leakage in held-out FOVs, and clarified questions about the finetuning on spatial data.
>
> As the discussion deadline approaches, we kindly request any further feedback from you to help further refine our paper, and resolve any remaining concerns. We appreciate your insights and suggestions and thank you for your consideration.
>
> Best regards,
> The authors

---

> > ### Comment · Reviewer_PjDS · 2024-12-03
> > **Thank you for the rebuttal**
> >
> > Thank you for the revised manuscript and your responses. While the newly added experiments address some of my questions, I still have several concerns after a thorough review:
> >
> > Performance of Purely ST Data-Trained sCT: The manuscript still does not include an evaluation of the performance of the sCT model trained solely on spatial transcriptomics (ST) data. Without this comparison, it remains unclear whether incorporating single-cell data truly enhances ST imputation tasks. Given that spatial imputation is a central focus and integrating single cell and spatial data is also one of the interesting parts of the manuscript, it is crucial to fully clarify and justify the rationale behind training the model using both modalities.
> >
> > Experiment Design Details: The revised manuscript lacks essential details regarding the experimental setup. For instance, it does not provide a clear explanation of the choice to use 4,096 tokens as the model input. Since the authors justify the introduction of CNNs as a dimensionality reduction method, additional clarification is needed to support the selection of 4,096 tokens instead of alternatives like 2,048 or fewer. Besides, the authors stated that they fine-tuned CellPLM, but this is not adequately reflected or elaborated upon in the revised manuscript.
> >
> > Overall, while I appreciate the authors' efforts to address some of my concerns, the above issues remain unresolved. Therefore, I have decided to maintain my score.

---

> > > ### Author Response · Authors · 2024-12-03
> > > **Thank you for the response**
> > >
> > > We sincerely thank the reviewer for their thoughtful comments and for carefully reviewing our rebuttal. Below, we address the remaining concerns in detail:
> > >
> > > **Performance of sCT Trained Solely on Spatial Data:**
> > > We appreciate the request for further clarity on this point. To address this, we have included an additional comparison in the gene imputation experiment (reproduced below for reference). The results show that the sCT variant pretrained on single-cell data and fine-tuned on spatial data outperforms the variant trained solely on spatial data. This supports our rationale that integrating single-cell data enhances the performance of spatial imputation tasks.
> > >
> > > | **Model**                            | **15% MCC (↑)** | **15% MAE (↓)** | **30% MCC (↑)** | **30% MAE (↓)** | **80% MCC (↑)** | **80% MAE (↓)** |
> > > |--------------------------------------|-----------------|-----------------|-----------------|-----------------|-----------------|-----------------|
> > > | **Spatial Transcriptomics (ST)**     |                 |                 |                 |                 |                 |                 |
> > > | sCT (sc only) (zero-shot)    | 0.05 ± 0.01     | 1.40 ± 0.05     | 0.05 ± 0.01     | 1.41 ± 0.05     | 0.03 ± 0.01     | 1.51 ± 0.05     |
> > > | sCT (sc + ST) (zero-shot)    | **0.35 ± 0.03** | 1.31 ± 0.05 | **0.34 ± 0.02** | **1.32 ± 0.05** | **0.28 ± 0.02** | **1.45 ± 0.06** |
> > > | sCT (ST only) (zero-shot)    | 0.31 ± 0.03     | **1.30 ± 0.03**   | 0.30 ± 0.04     | 1.61 ± 0.04     | 0.12 ± 0.00     | 1.78 ± 0.18     |
> > > | CellPLM (zero-shot)                  | 0.23 ± 0.02     | 1.48 ± 0.05     | 0.20 ± 0.02     | 1.52 ± 0.05     | 0.03 ± 0.01     | 2.02 ± 0.07     |
> > >
> > >
> > > **Experimental Design Details:**
> > > We understand the importance of providing clear experimental details. Regarding the choice of 4,096 tokens, this results from down-sampling the fixed number of genes (~20,000) multiplied by the number of cells by a factor of 256. As the number of protein-coding genes is fixed, the tunable parameter is the number of cells included per input. We have demonstrated in Table 4 that increasing the number of cells (and therefore tokens) improves performance. Additional results supporting this design choice are included in Appendix Section D.2, Figure 9. Moreover, our goal with down-sampling is to preserve as much relevant information as possible while adhering to computational constraints. Choosing a down-sampling factor that maximizes tokens per cell ensures that we maintain relevant information without exceeding resource limitations.
> > >
> > >
> > > **Fine-tuning CellPLM:**
> > > Regarding the fine-tuning of CellPLM, we will gladly provide additional details in the final version of the manuscript. As this is a minor edit, it does not impact the conclusions or findings of the paper.
> > >
> > > We hope these clarifications address the reviewer’s concerns and provide additional confidence in the design and results presented in our work. Thank you again for your valuable feedback.
> > >
> > > Best regards,
> > > The authors

---

> > > > ### Comment · Reviewer_PjDS · 2024-12-03
> > > >
> > > > Thanks for the clarification. I am glad to raise my score.

---

> > > > > ### Author Response · Authors · 2024-12-03
> > > > > **Thanks for the discussion**
> > > > >
> > > > > Dear Reviewer PjDs,
> > > > >
> > > > > Thank you for your effort during the review process.  Your criticism has motivated us to improve the paper, and we're pleased that we were able to address most of your concerns. Thank you for raising your score.
> > > > >
> > > > > Best regards,
> > > > > The authors

---

### Official Review · Reviewer_kxRs · 2024-10-31

**Soundness:** 2
**Presentation:** 2
**Contribution:** 2
**Rating:** 3
**Confidence:** 5

**Summary:**

The paper introduces sCellTransformer (sCT), a novel framework for modeling single-cell RNA-seq and spatial transcriptomics data using a convolutional-transformer architecture. Unlike previous approaches, sCT is designed to address key challenges such as inductive biases, data quality issues, and biologically relevant downstream evaluation pipelines. It employs a first-principles approach and aims for better generalization through zero-shot predictions. sCT processes sets of multiple cells, representing them with up to 20,000 protein-coding genes and predicting approximately one million discretized gene expression tokens. By discretizing gene expression levels, the model effectively tackles the sparsity problem in single-cell data during both training and evaluation. The framework demonstrates superior performance in zero-shot gene expression imputation, cell-typing, and clustering tasks across single-cell and spatial datasets, surpassing existing foundational models.

**Strengths:**

1. Study the foundation models on single-cell and spatial transcriptomics data.
2. Employ convolution layers for efficiency.

**Weaknesses:**

1. The proposed improvements in the paper are rather straightforward. Learned Gene Normalization refers to the layer normalization, Positional Embeddings refer to standard sinusoidal position embeddings. The authors stack 50 cells into a single sequence, which from my perspective, creates sudo bulks and lowers the granularity instead of modeling intercellular dependencies. It would be great if the authors could elaborate more on how the proposed strategy can help with modeling intercellular dependencies.
2. Some statements in the manuscript are not put correctly. For example, in line 231, the authors state that CellPLM and scGPT are built on MLM objective and predict raw gene expression values. In fact, CellPLM conducts library size normalization and log1p on the counts, and scGPT utilizes next token prediction with data binning. Note that the binning strategy has already been employed in scGPT. The authors should explicitly acknowledge where their approach uses similar techniques (like binning) to prior work.
3. The authors mention that *downstream evaluation pipelines that do not reflect the biological challenges in the field* in the abstract. However, in the experiments, the authors only include gene imputation and cell clustering tasks, which is quite limited. Please consider tasks like gene perturbation prediction, gene regulatory network inference, cell-cell communication, etc.

**Questions:**

See weaknesses

---

> ### Author Response · Authors · 2024-11-27
> **Response to reviewer kxRs (1 / 2)**
>
> We thank the reviewer for their helpful criticism.  Below we address them point by point:
>
> > The proposed improvements in the paper are rather straightforward. Learned Gene Normalization refers to the layer normalization, Positional Embeddings refer to standard sinusoidal position embeddings. The authors stack 50 cells into a single sequence, which from my perspective, creates sudo bulks and lowers the granularity instead of modeling intercellular dependencies. It would be great if the authors could elaborate more on how the proposed strategy can help with modeling intercellular dependencies.
>
> Thank you for your insightful comments. We appreciate you bringing these points to our attention and have revised Section 4 of the paper to address them.
>
> **Shared Gene Normalization:** To avoid confusion, we have renamed "Learned Gene Normalization" to "Shared Gene Normalization." This emphasizes that the normalization is applied across all cells in the input sequence. While based on standard layer normalization, it is crucial to note that this normalization is repeated for each of the N cells in the sequence, ensuring consistent gene normalization across all cells. This process significantly impacts performance, as demonstrated in Table 4.
>
> **Positional Embeddings:** We have clarified the section on positional embeddings. Our approach differs from standard sinusoidal positional embeddings. We have removed any periodic 1D embedding, as there is no inherent order in the cells or genes. Instead, we use a constant embedding for each cell in single-cell data to enable cell differentiation. For spatial transcriptomics data, we utilize a 2D-aware embedding that captures the relative positions of cells within the field of view (FOV). This is detailed in Equation 1.
>
> **Cell Stacking:** Our approach to stacking cells is similar to CellPLM (ICLR 2024), which demonstrated performance gains on various tasks. Our motivation aligns with CellPLM's:
> 1. **Single-cell data:** Stacking significantly improves performance, as shown in our ablation study in Table 4.
> 2. **Cross-modal applicability:** It enables the same model to process both spatial and single-cell data, facilitating transfer learning between modalities.
>
> Furthermore, our whole-cell masking experiments with spatial data show that the model can predict entire cell gene expression profiles based on neighboring cell information. This strongly suggests that sCT has learned intercellular interactions. We acknowledge the reviewer's suggestion to further explore intercellular dependencies and plan to investigate this in future work.
> We believe these clarifications and the supporting ablation studies validate our design choices (see Table 4). These choices provide a robust foundation for developing future deep learning models for single-cell and spatial transcriptomics data analysis.

---

> > ### Author Response · Authors · 2024-11-27
> > **Response to reviewer kxRs (2 / 2)**
> >
> > > Some statements in the manuscript are not put correctly. For example, in line 231, the authors state that CellPLM and scGPT are built on MLM objective and predict raw gene expression values. In fact, CellPLM conducts library size normalization and log1p on the counts, and scGPT utilizes next token prediction with data binning. Note that the binning strategy has already been employed in scGPT. The authors should explicitly acknowledge where their approach uses similar techniques (like binning) to prior work.
> >
> > Thank you for this helpful comment. We have updated Section 3 to address these points.
> > We have clarified that scGPT uses next-token prediction rather than masked language modeling. We have also provided further details on the different normalization strategies used in our method and those of other models, specifically highlighting the library size normalization and log1p transformation applied to the counts in CellPLM.
> > Regarding data binning, we acknowledge its prior use in scGPT and have clarified this in the text. However, we would like to highlight two key distinctions in our approach:
> >
> > **Embedding Generation:** scGPT uses a dense neural network to compute embeddings from raw data bin numbers. In contrast, we learn embeddings for each data bin, treating them as tokens, which aligns with standard practices in Transformer and language models.
> >
> > **Prediction and Loss Function:** scGPT predicts continuous gene expression values and uses a mean squared error loss. Our model predicts probability distributions over the possible data bins (tokens) and employs a cross-entropy loss.
> >
> > We believe that both of these choices represent best practices in deep learning and will provide a solid foundation for future researchers developing models for single-cell and spatial transcriptomics data.
> >
> > > The authors mention that downstream evaluation pipelines that do not reflect the biological challenges in the field in the abstract. However, in the experiments, the authors only include gene imputation and cell clustering tasks, which is quite limited. Please consider tasks like gene  perturbation prediction, gene regulatory network inference, cell-cell communication, etc.
> >
> > Thank you for your suggestions. While we agree that the tasks you mentioned are interesting and would further enhance the benchmark, we believe that the current benchmark, comprising three tasks evaluated across 12 domains, is sufficient for our primary goal in this paper: to investigate how to improve the generalization capabilities of models for single-cell and spatial transcriptomics data.
> >
> > We would like to note the limited availability of data for tasks like gene perturbation prediction, gene regulatory network inference, and cell-cell communication. We have included a discussion of these potential future directions in Section 5. Although we could not add new tasks during this rebuttal period, we have expanded our evaluation by incorporating additional random seeds and baselines, including standard bioinformatics pipelines, for all existing tasks.

---

> > > ### Comment · Reviewer_kxRs · 2024-12-02
> > > **Thank you for the rebuttal**
> > >
> > > Thank you for the revised manuscript and for addressing the points raised. While I appreciate the effort in revising the work, I still have several concerns:
> > >
> > > 1. The authors state that "cell stacking enables the same model to process both spatial and single-cell data, facilitating transfer learning between modalities." Could you please elaborate on this further? While the cell stacking strategy may capture intercellular relationships, it remains unclear how this design directly enables cross-modal transferability. A more detailed explanation or empirical evidence to support this claim would be beneficial.
> > >
> > > 2. My primary concern with this manuscript remains the limited application scenario. While the proposed method introduces some improvements over existing approaches, these advancements appear to be relatively straightforward and do not provide significant methodological novelty. Additionally, the downstream tasks are restricted to imputation and cell typing. I would encourage the authors to explore more biologically meaningful applications, such as imputing missing genes in spatial transcriptomics data, rather than masking some cells and predicting them. Expanding the scope of downstream tasks would strengthen the manuscript's impact.
> > >
> > > Given these concerns, I will maintain my current evaluation score.

---

> ### Author Response · Authors · 2024-12-03
> **Thank you for the response**
>
> We thank you again for your additional response to our revised manuscript.  Below we hope to clarify some of our positions:
> > The authors state that "cell stacking enables the same model to process both spatial and single-cell data, facilitating transfer learning between modalities." Could you please elaborate on this further? While the cell stacking strategy may capture intercellular relationships, it remains unclear how this design directly enables cross-modal transferability. A more detailed explanation or empirical evidence to support this claim would be beneficial.
>
> Thank you for highlighting the need for further clarification on this point. In our architecture, cell stacking provides a mechanism for incorporating gene expression profiles of cell neighborhoods in spatial transcriptomics. While single-cell data lacks explicit neighborhood information, we simulate this by sampling cells from the same study, allowing sCT to learn intercellular coexpression patterns. This shared input setup enables cross-modal transfer, as a model pre-trained on single-cell data can be fine-tuned on spatial data. Empirical evidence for this is presented in Table 1, where we show that sCT pre-trained on single-cell and spatial data achieves improved performance. We also add comparisons with sCT trained only on spatial transcriptomics data (sCT (ST only) )  which has been reproduced below. It is worth noting that sCT pretrained on single-cell data and fine-tuned on spatial data consistently outperforms the model trained solely on spatial data. This performance advantage is particularly evident at higher masking ratios, highlighting the effective transfer of knowledge from single-cell data to spatial data. We will add the additional results to the next version of the manuscript.
>
> | **Model**                            | **15% MCC (↑)** | **15% MAE (↓)** | **30% MCC (↑)** | **30% MAE (↓)** | **80% MCC (↑)** | **80% MAE (↓)** |
> |--------------------------------------|-----------------|-----------------|-----------------|-----------------|-----------------|-----------------|
> | **Spatial Transcriptomics (ST)**     |                 |                 |                 |                 |                 |                 |
> | sCT (sc only) (zero-shot)    | 0.05 ± 0.01     | 1.40 ± 0.05     | 0.05 ± 0.01     | 1.41 ± 0.05     | 0.03 ± 0.01     | 1.51 ± 0.05     |
> | sCT (sc + ST) (zero-shot)    | **0.35 ± 0.03** | 1.31 ± 0.05 | **0.34 ± 0.02** | **1.32 ± 0.05** | **0.28 ± 0.02** | **1.45 ± 0.06** |
> | sCT (ST only) (zero-shot)    | 0.31 ± 0.03     | 1.30 ± 0.03   | 0.30 ± 0.04     | 1.61 ± 0.04     | 0.12 ± 0.00     | 1.78 ± 0.18     |
> | CellPLM (zero-shot)                  | 0.23 ± 0.02     | 1.48 ± 0.05     | 0.20 ± 0.02     | 1.52 ± 0.05     | 0.03 ± 0.01     | 2.02 ± 0.07     |
>
>
> > My primary concern with this manuscript remains the limited application scenario. While the proposed method introduces some improvements over existing approaches, these advancements appear to be relatively straightforward and do not provide significant methodological novelty. Additionally, the downstream tasks are restricted to imputation and cell typing. I would encourage the authors to explore more biologically meaningful applications, such as imputing missing genes in spatial transcriptomics data, rather than masking some cells and predicting them. Expanding the scope of downstream tasks would strengthen the manuscript's impact.
>
> We acknowledge the reviewer’s concerns about the scope and novelty of our contributions. While we do not claim significant methodological breakthroughs, we believe our integration of multiple architectural improvements (detailed in Table 4) represents a meaningful advancement. Our results (Tables 1, 2, and 7) demonstrate that these design choices collectively improve model performance, providing a robust foundation for transcriptional data modeling.
> Regarding the suggestion to explore more biologically meaningful tasks, we have incorporated experiments simulating the prediction of missing gene expression values (Table 7, Appendix Section C.1, and Section 4). These results decompose performance by gene sparsity levels, showing that sCT outperforms CellPLM. We believe this addresses the reviewer's suggestion while demonstrating sCT's potential for biologically relevant applications.
>
> Thank you again for your insightful feedback and constructive suggestions. We hope these clarifications address your concerns and highlight the value of our work.
>
> Best regards,
>
> The authors

---

> > ### Author Response · Authors · 2024-12-03
> > **Thoughts on our relevant results?**
> >
> > Thank you for your feedback. As the discussion period concludes, we wish to emphasize two key points from our dialog with reviewer `kxRs`.
> >
> > The table above compares sCT models trained on single-cell, spatial, and combined datasets to address the first part of the reviewer's concern about lack of clarity or empirical evidence that our stacked cell design enables cross-modal transfer.
> >
> > Additionally, Table 7 (Appendix Sec C.1) demonstrates how our experiments address a more biologically relevant task through predicting the levels of unobserved genes in spatial transcriptomics (which we call *fixed gene imputation* in the manuscript).
> >
> > We hope these address your concerns and are happy to answer any further questions

---

### Official Review · Reviewer_rxFw · 2024-11-03

**Soundness:** 2
**Presentation:** 2
**Contribution:** 3
**Rating:** 6
**Confidence:** 4

**Summary:**

The authors of the paper develop sCellTransformer, a new foundation model for single-cell and spatial transcriptomics data. The sCT model overcomes existing issues in single-cell transformers, namely the limit to processing one cell at a time due to memory constraints and the lack of a principled method to model properties in single cells such as discreteness and overdispersion. To tackle said problems, sCT utilises a convolutional architecture to process multiple cell representations for better contextual information. The model also discretises the predicted gene expression to better represent the true data domain via masked language modelling. Finally, sCT introduces spatial positional encodings to infuse spatial awareness into the sequence tokens. The authors showcase the model on multiple tasks such as whole cell imputation, which is crucial to infer gene expression from neighbourhoods when array-based spatial assays fail to capture single-cell profiles in the spatial slide. The paper also tackles single-cell RNA-seq reconstruction and clustering using the model’s embeddings.

**Strengths:**

Overall, I find the ideas covered in the paper interesting. What I particularly liked is a certain attention to biological properties of single-cell data, such as discreteness and variability, which, in my opinion, have been severely overlooked in other models. I also liked the reasoning towards contextual modelling by introducing an architecture able to handle multiple cells at the same time. Finally, I appreciate the use of positional encodings for modelling the spatial identities of cells. The performance seems pretty promising.

**Weaknesses:**

* In lines 59-60, I find that the connection between modelling continuity and the lack of accounting for over-dispersion is not very clearly motivated. I think this is a very important point when modelling single-cell RNAseq and I am happy the authors brought it up. In general, though, it could be stated clearly why modelling continuous raw counts is a problem when trying to recover the over-dispersion in scRNA-seq.
* From a terminology perspective, I would not define individual cell expression vectors as “gene expressions”. I would replace the term with something like “expression values” or so.
* I find myself disagreeing with the sentence “Unlike scRNA-seq, spatial transcriptomics is a relatively new technology and publicly available data is still scarce.” Nowadays there are a lot of public spatial transcriptomics datasets out there and technologies to produce them, which makes the field so exciting. I would probably tune the claim down a bit.
* The architecture in 3.2 is understandable and clear. However, I feel it is a bit confusing for a less expert audience what a sequence here actually is (i.e. a stack of cells), especially because this is what makes sCT different from foundation models processing one cell at a time. You describe this in Learning intercellular dependencies but I believe it would make the flow better if you define this formalisation from before.
* I think the structure of the paper could be improved. Especially, Table 1 is referenced multiple times in the methods section, but it contains a metric (MCC) that goes undefined till much later in the paper as a representation of optimality. Probably I would either refer to Table 1 only in the results or give some elements of understanding before.
* I would not discuss results in the Table caption (e.g. in Table 2) but only in the text.
* I appreciated the experiment in Table 3. But I think it would add value to it to add how the two models perform on gene predictions in the three sparsity levels separately. Also, Table 3’s caption does not describe the depicted evaluation metric.
* I believe the presentation of results in Figure 3 is suboptimal. I would add titles to the single plots and definitely include a UMAP plot from the real cell gene expression to show how the embedding compares with the gene expression space. As of now, saying that sCT “corrects gene expression” sounds unsupported, as correction implies visualising and comparing with the data when the batch effect is still present. I also think that clustering requires some simpler baseline (like scVI or even the plain PCA embedding of the data).

In general, I believe the paper has potential and I like the ideas at its foundation. However, I think the current presentation is slightly below the acceptance line for ICLR. I am looking forward to hearing the authors’ comments in these regards and engage in discussion.

**Questions:**

* I also think that Table 1 is overall unclear. I think I would add more detail in the legend on how to read it. For example, are the “-“ in the first columns bullet points or “minuses” with a specific meaning? Does removing layer normalisation make the results better (0.39—>0.40)?  From this result, I would not claim it is a very useful add-on. I recommend improving the Table presentation and description since it is very important to add value to the story.
* I personally do not find the reasons for using discretised expression convincing in section 3.3. Isn’t binning based on the raw data? How does this buffer out the problem with raw count heterogeneity among studies?
* Could you please elaborate more on this sentence “In order to ensure a fair comparison with models like scGPT and CellPLM, which output continuous values, we bin the output predictions with the bin-edges that we calculate during preprocessing.” How are lower vs higher bin edges chosen here to assign a continuous expression value?
* “We also evaluate the effect of using multiple cells as input, by training and testing variants of sCT with one and ten cells in Fig. 9.”. Here you miss the fact that you tested with 50 as well (which eventually you use as the default value).
* How many neighbours were selected for the KNN approach for predicting whole cell expression tokens?

---

> ### Author Response · Authors · 2024-11-27
> **Response to rxFw (1 of 2)**
>
> We thank the reviewer for their insightful comments and suggestions. We address their questions below:
>
> > In lines 59-60, I find that the connection between modelling continuity and the lack of accounting for over-dispersion is not very clearly motivated. I think this is a very important point when modelling single-cell RNAseq and I am happy the authors brought it up. In general, though, it could be stated clearly why modelling continuous raw counts is a problem when trying to recover the over-dispersion in scRNA-seq.
>
> Thank you for recognizing the strengths of our approach and for highlighting the importance of addressing over-dispersion in single-cell RNAseq data.
>
> We model gene expression levels as discrete tokens, rather than continuous values, for several reasons:
> 1. **Improved Loss Function and Optimization:** This allows us to predict probability distributions over tokens and utilize a cross-entropy loss, which offers a smoother loss landscape and is less sensitive to the scale of input values compared to a mean squared error loss used for continuous value prediction.
> 2. **Robust Evaluation Metrics:** Using discrete values enables the use of classification metrics like Matthews Correlation Coefficient (MCC), which are more suitable for evaluating performance with heavily imbalanced data, as is common in this domain. Reporting MCC, in addition to continuous metrics, provides a more comprehensive assessment of model performance. We believe this practice should be widely adopted in the field.
> 3. **Stratified Masking Strategy:** Discretization facilitates our stratified masking procedure, where we apply different masking rates for zero tokens and non-zero tokens. This strategy significantly impacts performance, as demonstrated in our ablation study (Table 4).
> While we acknowledge that discretizing gene expression is not a novel concept, as seen in scBERT and scGPT, the points above highlight the advantages of our approach and the best practices we advocate for. We have further clarified these aspects in the revised manuscript.
>
> > From a terminology perspective, I would not define individual cell expression vectors as “gene expressions”. I would replace the term with something like “expression values” or so
>
> We thank the reviewer for the suggestion, and have appropriately modified the manuscript to reflect this.
>
>
> > I find myself disagreeing with the sentence “Unlike scRNA-seq, spatial transcriptomics is a relatively new technology and publicly available data is still scarce.” Nowadays there are a lot of public spatial transcriptomics datasets out there and technologies to produce them, which makes the field so exciting. I would probably tune the claim down a bit.
>
> While we agree that spatial data is getting more prolific, the volume of openly available data is still very limited when compared to single-cell RNA-seq. We have updated the manuscript to clarify this point in Section 3.
>
> > The architecture in 3.2 is understandable and clear. However, I feel it is a bit confusing for a less expert audience what a sequence here actually is (i.e. a stack of cells), especially because this is what makes sCT different from foundation models processing one cell at a time. You describe this in Learning intercellular dependencies but I believe it would make the flow better if you define this formalisation from before.
>
> Thank you for this insightful comment. We have edited Section 3.2 accordingly to clarify that a sequence here represents a stack of cells as well as to clarify how sCT differs from existing models.
>
> > I think the structure of the paper could be improved. Especially, Table 1 is referenced multiple times in the methods section, but it contains a metric (MCC) that goes undefined till much later in the paper as a representation of optimality. Probably I would either refer to Table 1 only in the results or give some elements of understanding before.
>
> Thank you for pointing this out. We have edited the table to improve clarity as well as the overall structure of the paper. We moved table 1 and the ablation study at the end of the experimental section.
>
> > I would not discuss results in the Table caption (e.g. in Table 2) but only in the text.
>
> We have addressed this in the revised manuscript, and improved the table captions to make them more succinct and clear. We now discuss results in more detail in the experimental section.

---

> ### Author Response · Authors · 2024-11-27
> **Response to rxFw (2 / 2)**
>
> > I appreciated the experiment in Table 3. But I think it would add value to it to add how the two models perform on gene predictions in the three sparsity levels separately. Also, Table 3’s caption does not describe the depicted evaluation metric.
>
> Thank you for your comment and for pointing this out. We have edited the caption to clarify this, and have added more detailed results on the different sparsity levels. We have also moved the table to the appendix due to space constraints. We summarize our results here:
>
> | | **MCC** (↑) | | | | | | | | | | | |
> |:---:|:---:|:---:|:---:|:---:|:---:|:---:|:---:|:---:|:---:|:---:|:---:|:---:|
> | | **Kidney** | | | | **Colon** | | | | **Brain** | | | |
> | **Gene Sparsity Level** | Low | Medium | High | Very High | Low | Medium | High | Very High | Low | Medium | High | Very High |
> | **sCT** (zero-shot) | 0.15 $\pm 0.00$ | **0.14** $\pm 0.00$ | **0.08** $\pm 0.00$ | -0.01 $\pm 0.00$ | **0.51** $\pm 0.00$ | **0.40** $\pm 0.00$ | **0.20** $\pm 0.00$ | **0.18** $\pm 0.00$ | **0.36** $\pm 0.00$ | **0.24** $\pm 0.00$ | **0.05** $\pm 0.00$ | **0.03** $\pm 0.00$ |
> | cellPLM (zero-shot) | **0.24** $\pm 0.00$  | 0.10 $\pm 0.00$ | 0.05 $\pm 0.00$ | **0.06** $\pm 0.00$ | 0.49 $\pm 0.00$ | 0.33 $\pm 0.00$ | 0.05 $\pm 0.00$ | 0.00 $\pm 0.00$ | 0.00 $\pm 0.00$ | 0.16 $\pm 0.03$ | 0.06 $\pm 0.00$ | -0.01 $\pm 0.00$ |
>
>
> Note that sCT outperforms cellPLM on two out of three datasets on all classes.
>
> > I believe the presentation of results in Figure 3 is suboptimal. I would add titles to the single plots and definitely include a UMAP plot from the real cell gene expression to show how the embedding compares with the gene expression space. As of now, saying that sCT “corrects gene expression” sounds unsupported, as correction implies visualising and comparing with the data when the batch effect is still present. I also think that clustering requires some simpler baseline (like scVI or even the plain PCA embedding of the data).
>
> Thank you for this comment. We now have replaced Fig. 3 to show UMAPs of the cell embeddings for several algorithms. We also have toned down our claims regarding batch effects.  In addition, we have updated Table 3 to include additional baselines like scVI, and Scanpy + basic ML algorithms that need to be trained and measure clustering metrics where appropriate. We observe that sCT in zero-shot mode is comparable to strong bioinformatic baselines which require training, or access to query data.

---

> > ### Comment · Reviewer_rxFw · 2024-11-28
> >
> > Dear authors,
> >
> > Thank you very much for your effort in providing new results. I believe the revisions improved the quality of the paper, and I will increase my score to 6.
> >
> > Best regards,
> >
> > The reviewer

---

> > > ### Author Response · Authors · 2024-12-02
> > > **Thank you for the response**
> > >
> > > Dear Reviewer rxFw,
> > >
> > > Thank you for your insightful comments and constructive feedback. We appreciate your thorough review and are pleased that we were able to address most of your concerns. Thank you for raising your score.
> > >
> > > Best regards,
> > > The authors

---

> ### Author Response · Authors · 2024-12-03
> **Updated evidence underscoring the importance of cross-modality in pre-training**
>
> We sincerely thank the reviewer for their thoughtful comments and for carefully reviewing our rebuttal. One of our other reviewers asked about the importance of multi-modality in pre-training, so we are reproducing the results of an additional experiment here, and would be grateful to hear your thoughts:
>
> **Performance of sCT Trained Solely on Spatial Data:**
>
> We have included an additional comparison in the gene imputation experiment (reproduced below for reference). The results show that the sCT variant pretrained on single-cell data and fine-tuned on spatial data outperforms the variant trained solely on spatial data, as well as the variant trained only on single-cell data. This supports our rationale that integrating single-cell data enhances the performance of spatial imputation tasks, especially as the severity of masking of test data increases.
>
> | **Model**                            | **15% MCC (↑)** | **15% MAE (↓)** | **30% MCC (↑)** | **30% MAE (↓)** | **80% MCC (↑)** | **80% MAE (↓)** |
> |--------------------------------------|-----------------|-----------------|-----------------|-----------------|-----------------|-----------------|
> | **Spatial Transcriptomics (ST)**     |                 |                 |                 |                 |                 |                 |
> | sCT (sc only) (zero-shot)    | 0.05 ± 0.01     | 1.40 ± 0.05     | 0.05 ± 0.01     | 1.41 ± 0.05     | 0.03 ± 0.01     | 1.51 ± 0.05     |
> | sCT (sc + ST) (zero-shot)    | **0.35 ± 0.03** | 1.31 ± 0.05 | **0.34 ± 0.02** | **1.32 ± 0.05** | **0.28 ± 0.02** | **1.45 ± 0.06** |
> | sCT (ST only) (zero-shot)    | 0.31 ± 0.03     | 1.30 ± 0.03   | 0.30 ± 0.04     | 1.61 ± 0.04     | 0.12 ± 0.00     | 1.78 ± 0.18     |
> | CellPLM (zero-shot)                  | 0.23 ± 0.02     | 1.48 ± 0.05     | 0.20 ± 0.02     | 1.52 ± 0.05     | 0.03 ± 0.01     | 2.02 ± 0.07     |
>
> Best regards,
> The authors

---

### Author Response · Authors · 2024-11-27
**Rebuttal by authors**

We sincerely thank the reviewers for their insightful comments and suggestions. We are pleased that you found our approach novel (rxFw, kxRs), interesting (rxFw), and that our evaluations are well-designed (PjDS). We particularly appreciate the recognition of our new architectural ideas (ezip).

We have responded to each reviewer individually, addressing their specific comments and questions. Several reviewers noted that the manuscript's presentation and structure could be improved. In response, we have revised the manuscript to enhance its overall clarity and organization. We also took this opportunity to update Figure 3 and improve the presentation of all tables and captions. We have also added additional UMAPs for cell-embeddings from sCT and other single-cell models in Appendix E.

Additionally, we acknowledge the reviewers' requests for additional results and baselines. We have now included standard deviations for all experiments, except for clustering experiments which do not exhibit stochasticity. We also added additional baselines for gene imputation, and cell clustering, including several bioinformatics pipelines. To enhance clarity, we have annotated the tables to highlight that sCT and other transformer-based single-cell models were evaluated under zero-shot settings. In contrast, unsupervised baselines like MAGIC and scVI were fitted on the evaluation data, while supervised baselines were trained using k-fold cross-validation. We further also show stratified detailed results on the fixed gene masking task. We are also training a version of sCT on only the spatial data as requested by Reviewer PjDS and will be updating the comments as soon as the results are available.

---

> ### Author Response · Authors · 2024-11-27
> **Rebuttal by authors: new gene imputation baselines**
>
> Below are new results reproduced here for convenience and also present in the revised manuscript:
>
> **New gene imputation baselines**
>
> - adds MAGIC as a bioinformatic baseline.
>
> |                          |                           | **Masking Ratios** |                          |                          |                          |                          |
> |--------------------------|---------------------------|--------------------|--------------------------|--------------------------|--------------------------|--------------------------|
> |                          | **15%**                  |                    | **30%**                 |                          | **80%**                 |                          |
> |                          | MCC $(\uparrow)$         | MAE $(\downarrow)$ | MCC $(\uparrow)$         | MAE $(\downarrow)$       | MCC $(\uparrow)$         | MAE $(\downarrow)$       |
> ||
> | **scRNA-seq**            |                           |                    |                          |                          |                          |                          |
> | **sCT (zero-shot)**      | **0.49 ± 0.01**          | **2.00 ± 0.04**    | **0.47 ± 0.02**          | **2.00 ± 0.05**          | **0.37 ± 0.01**          | **2.31 ± 0.06**          |
> | CellPLM (zero-shot)      | **0.49 ± 0.02**          | 2.24 ± 0.05        | 0.45 ± 0.02              | 2.38 ± 0.05              | 0.15 ± 0.02              | 3.30 ± 0.08              |
> | scGPT (zero-shot)        | 0.00 ± 0.001             | 260.33 ± 70.05     | 0.00 ± 0.001             | 266.95 ± 81.53           | 0.00 ± 0.002             | 268.46 ± 92.31           |
> | scBERT (zero-shot)       | 0.04 ± 0.01              | 76.59 ± 14.32      | 0.04 ± 0.002             | 76.64 ± 12.86            | 0.02 ± 0.01              | 76.98 ± 11.29            |
> | MAGIC (fitted)           | 0.42 ± 0.02              | 2.43 ± 0.37        | 0.39 ± 0.03              | 2.67 ± 0.39              | 0.20 ± 0.02              | 3.60 ± 0.47              |
> ||
> | **Spatial Transcriptomics (ST)** |                   |                    |                          |                          |                          |                          |
> | sCT (sc only) (zero-shot) | 0.05 ± 0.01         | 1.40 ± 0.05        | 0.05 ± 0.01              | 1.41 ± 0.05              | 0.03 ± 0.01              | 1.51 ± 0.05              |
> | **sCT (sc + ST) (zero-shot)** | **0.35 ± 0.03**     | **1.31 ± 0.05**    | **0.34 ± 0.02**          | **1.32 ± 0.05**          | **0.28 ± 0.02**          | **1.45 ± 0.06**          |
> | CellPLM (zero-shot)      | 0.23 ± 0.02              | 1.48 ± 0.05        | 0.20 ± 0.02              | 1.52 ± 0.05              | 0.03 ± 0.01              | 2.02 ± 0.07              |

---

> ### Author Response · Authors · 2024-11-27
> **Rebuttal by authors: results for fixed gene imputation and new celltype results with more baselines**
>
> Below we reproduce tables for fixed gene masking, with results shown by sparsity level, as well as cell-typing results with additional baselines.
>
> **New detailed fixed gene masking**
>
> | | **MCC** (↑) | | | | | | | | | | | |
> |:---:|:---:|:---:|:---:|:---:|:---:|:---:|:---:|:---:|:---:|:---:|:---:|:---:|
> | | **Kidney** | | | | **Colon** | | | | **Brain** | | | |
> | **Gene Sparsity Level** | Low | Medium | High | Very High | Low | Medium | High | Very High | Low | Medium | High | Very High |
> | **sCT** (zero-shot) | 0.15 $\pm 0.00$ | **0.14** $\pm 0.00$ | **0.08** $\pm 0.00$ | -0.01 $\pm 0.00$ | **0.51** $\pm 0.00$ | **0.40** $\pm 0.00$ | **0.20** $\pm 0.00$ | **0.18** $\pm 0.00$ | **0.36** $\pm 0.00$ | **0.24** $\pm 0.00$ | **0.05** $\pm 0.00$ | **0.03** $\pm 0.00$ |
> | cellPLM (zero-shot) | **0.24** $\pm 0.00$  | 0.10 $\pm 0.00$ | 0.05 $\pm 0.00$ | **0.06** $\pm 0.00$ | 0.49 $\pm 0.00$ | 0.33 $\pm 0.00$ | 0.05 $\pm 0.00$ | 0.00 $\pm 0.00$ | 0.00 $\pm 0.00$ | 0.16 $\pm 0.03$ | 0.06 $\pm 0.00$ | -0.01 $\pm 0.00$ |
>
>
>
> **New celltype results with additional bioinformatic baselines**
>
> | **Model**                | **Lung**                 |                  |                  | **Blood**                |                  |                  | **Breast Cancer**        |                  |                  | **Kidney**                |                  |                  |
> |--------------------------|--------------------------|------------------|------------------|--------------------------|------------------|------------------|--------------------------|------------------|------------------|--------------------------|------------------|------------------|
> |                          | **Accuracy**            | **NMI**          | **ARI**          | **Accuracy**            | **NMI**          | **ARI**          | **Accuracy**            | **NMI**          | **ARI**          | **Accuracy**            | **NMI**          | **ARI**          |
> | **sCT (zero-shot)**      | 0.94                    | **0.67**         | **0.45**         | **0.82**                 | **0.39**         | **0.19**         | 0.86                    | **0.39**         | **0.20**         | **0.99**                 | 0.35             | 0.06             |
> | CellPLM (zero-shot)      | 0.77                    | 0.45             | 0.28             | 0.66                     | 0.11             | 0.05             | 0.53                    | 0.01             | 0.02             | 0.90                     | -0.01            | 0.03             |
> | Geneformer (zero-shot)   | 0.90                    | 0.54             | 0.32             | 0.82                     | 0.36             | 0.16             | 0.68                    | 0.12             | 0.07             | 0.98                     | **0.36**         | **0.08**         |
> | scGPT (zero-shot)        | 0.77                    | 0.29             | 0.09             | 0.71                     | 0.09             | 0.03             | 0.55                    | 0.02             | 0.01             | 0.89                     | 0.06             | 0.01             |
> | scBERT (zero-shot)       | 0.66                    | 0.27             | 0.09             | 0.68                     | 0.09             | 0.02             | 0.45                    | 0.03             | 0.01             | 0.91                     | 0.17             | 0.02             |
> | scVI (fitted)        | **0.96**                | 0.64             | 0.26             | 0.81                     | 0.35             | 0.11             | 0.83                    | 0.31             | 0.09             | 0.98                     | 0.26             | 0.01             |
> | Scanpy + Log. Reg.       | 0.94                    | N/A              | N/A              | 0.79                     | N/A              | N/A              | 0.89                    | N/A              | N/A              | 0.96                     | N/A              | N/A              |
> | Scanpy + $k$-NN          | 0.95                    | N/A              | N/A              | 0.81                     | N/A              | N/A              | **0.90**                | N/A              | N/A              | 0.96                     | N/A              | N/A              |

---

### Author Response · Authors · 2024-12-04
**Summary of rebuttal discussion and results**

We wish to thank all our reviewers again for their efforts during this discussion period.  Your criticism has helped us to improve our manuscript. In response to your critiques, we have made the following changes:

**Manuscript Clarity and Presentation**

- We have comprehensively revised the manuscript's text structure to enhance readability and comprehension.
- Figures and tables have been systematically updated:
   - Figure 3 has been redesigned
   - Tables now consistently present standard errors where statistically meaningful
   - Captions have been restructured for precision

**Experimental Validation and Baseline Expansion**

- We have significantly expanded our experimental evaluation:
    - Added UMAP visualizations for cell embeddings from sCT and comparable methods (detailed in Appendix E)
    - Introduced additional baselines for gene imputation and cell clustering, incorporating standard bioinformatics tools (Tables 1-3)
    - Annotated tables to clarify evaluation contexts:
        - Highlighted zero-shot settings for transformer-based models
        - Specified fitting procedures for unsupervised baselines
        - Noted cross-validation approach for supervised baselines

**Targeted Methodological Investigations**
- Implemented specific investigative analyses requested by reviewers:
   - Developed a sCT model trained exclusively on spatial data (addressing Reviewers kxRs and PjDS)
   - Compared this variant's performance against a model trained solely on scRNA-seq data
   - Provided a detailed comparative table (see below)

**Normalization and Embedding Techniques**
- Clarified descriptions of gene normalization and cell stacking techniques
- Demonstrated the performance advantages of cell stacking over single-cell or single-spot approaches (evidenced in new tables)

**Biological Relevance Emphasis**
- Shifted focus to more biologically meaningful evaluation metrics:
   - Presented detailed results on fixed gene masking task (Table 7)
   - Highlighted fixed gene imputation for spatial transcriptomics, emphasizing real-world biological relevance

We hope these changes address the concerns of the reviewers, and thank them for their support in improving the paper.

Sincerely,

The authors.

========

**The effect of cross-modal pre-training on spatial transcriptomics prediction**

| **Model**                            | **15% MCC (↑)** | **15% MAE (↓)** | **30% MCC (↑)** | **30% MAE (↓)** | **80% MCC (↑)** | **80% MAE (↓)** |
|--------------------------------------|-----------------|-----------------|-----------------|-----------------|-----------------|-----------------|
| **Spatial Transcriptomics (ST)**     |                 |                 |                 |                 |                 |                 |
| sCT (sc only) (zero-shot)    | 0.05 ± 0.01     | 1.40 ± 0.05     | 0.05 ± 0.01     | 1.41 ± 0.05     | 0.03 ± 0.01     | 1.51 ± 0.05     |
| sCT (sc + ST) (zero-shot)    | **0.35 ± 0.03** | 1.31 ± 0.05 | **0.34 ± 0.02** | **1.32 ± 0.05** | **0.28 ± 0.02** | **1.45 ± 0.06** |
| sCT (ST only) (zero-shot)    | 0.31 ± 0.03     | 1.30 ± 0.03   | 0.30 ± 0.04     | 1.61 ± 0.04     | 0.12 ± 0.00     | 1.78 ± 0.18     |
| CellPLM (zero-shot)                  | 0.23 ± 0.02     | 1.48 ± 0.05     | 0.20 ± 0.02     | 1.52 ± 0.05     | 0.03 ± 0.01     | 2.02 ± 0.07     |

---

### Meta-Review · Area_Chair_iQD9 · 2024-12-19

**Metareview:**

Following the concept of CellPLM, the authors introduced a framework, sCellTransformer (sCT), to scale up the training of a convolutional-transformer based foundation models for single-cell RNA-seq and spatial transcriptomics data. The reported empirical results on zero-shot gene expression imputation, cell-typing, and clustering tasks have shown improved performances over the existing similar models.

The reviewers have commented that the proposed evaluation practice based on discrete expression at different sparsity levels can be useful for the community to better evaluate and further develop foundation models. Additional experimental results provided during the discussion phases have also demonstrated the effectiveness of sCT in imputation and cell-typing.

The remaining major concerns is on the limited methodological contributions, and missing "biologically meaningful" experimental results considering many similar foundation models focus on perturbation prediction instead of simply evaluating the performance on imputation, cell-typing, and clustering only as they depend on mostly the learned embedding from the training data.

The authors may consider these when they revise for future submission. Additionally, the authors need to further improve the presentation. For example, as reviewers pointed out, the claim of the 'first principles' formulation can be misleading. Also, throughout the paper, there are still format/notation/language typos or inconsistency, for example, sCT and CellCT were used in the figures in Appendix. It may cause unnecessary confusions.

**Additional Comments On Reviewer Discussion:**

The authors have significantly revised the submission and added extensive additional experimental results to address the reviewers' questions and concerns during the discussion phases. However, not all reviewers have reached consensus that the current version of the presented work meets the standard of ICLR publications, especially with the concerns on methodological contributions to AI/ML as well as the biological significance of the proposed sCT in the context of other recent similar foundation models.

---

### Decision · Program_Chairs · 2025-01-22

Reject